# QUERY-EFFICIENT ZEROTH-ORDER ALGORITHMS FOR NONCONVEX OPTIMIZATION

## ABSTRACT

Zeroth-order optimization (ZO) has been a powerful framework for solving black-box problems, which estimates gradients using zeroth-order data to update variables iteratively. The practical applicability of ZO critically depends on the efficiency of single-step gradient estimation and the overall query complexity. However, existing ZO algorithms cannot achieve efficiency on both simultaneously. In this work, we consider a general constrained optimization model with black-box objective and constraint functions. To solve it, we propose novel algorithms that can achieve the state-of-the-art overall query complexity bound of $\mathcal{O}(d/\epsilon^4)$ to find an $\epsilon$-stationary solution ($d$ is the dimension of variable space), while reducing the queries for estimating a single-step gradient from $\mathcal{O}(d)$ to $\mathcal{O}(1)$. Specifically, we integrate block updates with gradient descent ascent and a block gradient estimator, which leads to two algorithms, ZOB-GDA and ZOB-SGDA, respectively. Instead of constructing full gradients, they estimate only partial gradients along random blocks of dimensions, where the adjustable block sizes enable high single-step efficiency without sacrificing convergence guarantees. Our theoretical results establish the finite-sample convergence of the proposed algorithms for nonconvex optimization. Finally, numerical experiments on a practical problem demonstrate that our algorithms require over ten times fewer queries than existing methods.

## 1 INTRODUCTION

In practical problems, it is common to encounter real systems that lack analytical expressions or models. In such cases, only zeroth-order (input-output) information of the systems is accessible. The lack of higher-order information makes it especially difficult to optimize these systems. In this research, we consider a general constrained optimization model for these problems:

$$\min_{x \in \mathbb{R}^{d_x}} h(x) \qquad \text{s.t. } c_j(x) \le 0, \ \forall j \in \mathcal{J}, \tag{1}$$

where $h : \mathbb{R}^{d_x} \to \mathbb{R}$ is the objective function and each $c_j : \mathbb{R}^{d_x} \to \mathbb{R}, \forall j \in \mathcal{J}$ is a constraint function. Both $h(x)$ and $c_j(x)$ do not have analytical expressions and are treated as black boxes, i.e., only the input $x$ and the corresponding deterministic function outputs $h(x)$ or $c_j(x)$ are observable. Neither $h$ nor $c_j, \forall j \in \mathcal{J}$ is necessarily convex.

Problems in the form of (1) arise across many domains, such as power systems (Hu et al., 2024; Zhou et al., 2025), simulation optimization (Park & Kim, 2015), and machine learning (Nguyen & Balasubramanian, 2023). However, traditional model-based or gradient-based algorithms are inapplicable to problem (1), as they rely on first-order or second-order information (e.g., gradients or Hessians) of $h(x)$ and $c_j(x)$, which is not available. Zeroth-order optimization (ZO), a representative method in derivative-free optimization, offers a promising approach to this type of optimization problem, and has been broadly applied (Fu et al., 2015; Liu et al., 2020a; Malladi et al., 2023; Lam & Zhang, 2024). The fundamental idea behind ZO is to construct estimators of first-order information using zeroth-order data (Berahas et al., 2022), and integrate these estimators into gradient-based algorithms, such as gradient descent, to seek optimal or high-quality solutions.

Under the iterative ZO framework, the efficiency of single-step gradient estimation and overall query complexity jointly determine the practical applicability of ZO (Scheinberg, 2022). They refer to the number of function values required to generate a single-step gradient and a final solution, respec-

tively. The traditional *coordinate-wise gradient estimation* (CGE) requires estimating partial gradients along all dimensions separately based on finite differences of function values (Kiefer & Wolfowitz, 1952). Although CGE-based algorithms can generally enjoy state-of-the-art overall query complexities due to the controllable bias and variance of CGEs (Xu et al., 2024; Zhou et al., 2025), the requirement of $\mathcal{O}(d)$ queries for estimating a single-step gradient makes them inefficient for high-dimensional problems ($d$ is the dimension of variable space). In contrast, the prevalent *randomized gradient estimation* (RGE) only requires one or two function values to construct a gradient estimator along a random direction (Flaxman et al., 2005; Nesterov & Spokoiny, 2017). RGE-based algorithms have demonstrated excellent performance in unconstrained problems. However, they suffer from slow convergence when applied to constrained cases (such as (1)) due to large variances of gradient estimation (Liu et al., 2020a). This significantly limits their practical performance. In view of this dilemma, a fundamental question arises:

> *To solve (1), can we design zeroth-order algorithms that are query-efficient regarding both single-step and overall complexities?*

In this paper, we provide a positive answer to this question. We will utilize the framework of random block updates to design novel and query-efficient ZO algorithms for solving problem (1), and show that the proposed algorithms enjoy controllable single-step efficiency and the best-known overall query complexities. For a detailed discussion on related work, please refer to Appendix A.

## 1.1 MAIN CONTRIBUTIONS

We assume simultaneous zeroth-order access to $h(x)$ and $c_j(x), \forall j \in \mathcal{J}$ (i.e., we can observe all the function evaluations of $h(x)$ and $c_j(x), \forall j \in \mathcal{J}$ simultaneously via querying a $x$) but no gradient information. To handle the black-box constraints in (1), we adopt a primal-dual framework by reformulating it as a deterministic min-max problem:

$$\min_{x \in \mathbb{R}^{d_x}} \max_{y \in \mathcal{Y}} f(x,y), \tag{2}$$

where $f(x,y) = h(x) + y^{\mathrm{T}} c(x)$ is the Lagrange function of problem (1). Wherein, $c(x) = (c_1(x), \cdots, c_{d_y}(x))^{\mathrm{T}}$ with $d_y = |\mathcal{J}|$; $\mathcal{Y} = \{y \in \mathbb{R}^{d_y} | y \geq 0\}$ is the feasible set of Lagrange multiplier. Clearly, $f(x,y)$ is *nonconvex-concave*, i.e., nonconvex in $x$ and concave in $y$, when $h(x)$ and $c_j(x), \forall j \in \mathcal{J}$ are not assumed convex. Then, solving problem (2) can provide optimal or high-quality solutions to problem (1) (Nesterov et al., 2018). The detailed contributions of this work are summarized as follows.

Table 1: Comparison of single-step and overall query complexities

| Algorithms | Gradient Estimator | Queries per Step | Overall Queries |
|---|---|---|---|
| SZO-ConEX (Nguyen & Balasubramanian, 2023) | RGE | $\mathcal{O}(1)$ | $\mathcal{O}(d/\epsilon^6)$ |
| ZOAGP (Xu et al., 2024) | CGE | $\mathcal{O}(d)$ | $\mathcal{O}(d/\epsilon^4)$ |
| ZOB-GDA (Ours) | BCGE | $\mathcal{O}(b)$ | $\mathcal{O}(d/\epsilon^6)$ |
| ZOB-SGDA (Ours) | BCGE | $\mathcal{O}(b)$ | $\mathcal{O}(d/\epsilon^4)$ |

Note: In our algorithms, $d = d_x$. $b$ is the block size that can be chosen from $\{1, 2, \cdots, d\}$. Overall queries refer to the number of queries required to achieve an $\epsilon$-stationary/KKT point.

**Leveraging Block Updates with Zeroth-Order Algorithms to Solve (1).** In this research, we adopt the widely-used gradient descent ascent (GDA) framework to solve problem (1). However, directly applying RGE or CGE in GDA cannot exhibit satisfactory performance in both single-step and overall complexity (see our detailed discussion in Section 2.2). To address this, we combine the framework of block updates with GDA and smoothed GDA to develop two novel algorithms, called *zeroth-order block gradient descent ascent* (ZOB-GDA) and *zeroth-order block smoothed gradient descent ascent* (ZOB-SGDA). Rather than estimating a full gradient at each step, they randomly select a block of coordinates and update the variables using *block coordinate-wise gradient estimations* (BCGEs). The adoption of the BCGEs effectively controls the bias and variance of gradient estimations to be negligible and thereby accelerates convergence. Moreover, the block size is adjustable to control the number of queries required to construct a single-step gradient.

**Best-Known Query Complexities with Controllable Single-Step Efficiency.** We establish finite-sample guarantees for the proposed algorithms by analyzing the min-max problems (2) in nonconvex-concave settings. The query complexity results are summarized in Table 1 and compared with two representative algorithms. Specifically, ZOB-GDA can find an $\epsilon$-stationary point of $f(x,y)$ with a query complexity bound $\mathcal{O}(d/\epsilon^6)$, which differs from the bound for first-order GDA only by a factor of $d$. Moreover, ZOB-SGDA is shown to have the query complexity bound $\mathcal{O}(d/\epsilon^4)$, which aligns with the best-known results for solving deterministic nonconvex-concave problems. Different from existing methods, our algorithms also benefit from controllable efficiency in single-step gradient estimation, which makes them query-efficient for both single-step and overall complexities. The numerical results demonstrate that our algorithms can require over 10 times fewer queries for both a single step and overall complexity compared to existing methods.

## 2 PRELIMINARIES

**Notations.** For a positive integer $n$, we denote $[n] := \{1, 2, \cdots, n\}$. For a vector $x \in \mathbb{R}^{d_x}$, denote $x(i)$ as its $i$th entry. For a differentiable function $h(x) : \mathbb{R}^{d_x} \to \mathbb{R}$, denote $\nabla h(x)$ as its gradient at $x$ and $\nabla_i h(x), i \in [d_x]$ as the partial gradient along the $i$th dimension. Similarly, for a differentiable function $f(x,y) : \mathbb{R}^{d_x} \times \mathbb{R}^{d_y} \to \mathbb{R}$, denote the partial gradient w.r.t. $x$ (and $y$) by $\nabla_x f(x,y)$ (and $\nabla_y f(x,y)$). Without further specification, $\|\cdot\|$ denotes the $\ell_2$-norm in Euclidean space. The Euclidean projection operator onto a closed convex set $\mathcal{X}$ is denoted by $\mathcal{P}_{\mathcal{X}}[\cdot]$.

### 2.1 ASSUMPTIONS AND STATIONARITY MEASURE

Below, we present the key assumptions for our analysis and introduce the definition of stationarity measure for evaluating our proposed algorithms.

**Assumption 2.1.** The set $\mathcal{Y}$ is compact, i.e., $\mathcal{Y} := \{y \in \mathbb{R}^{d_y} | 0 \le y \le \overline{y}\}$ for some bounded $\overline{y} \in \mathbb{R}^{d_y}$. Moreover, $\Phi(x) = \max_{y \in \mathcal{Y}} f(x,y)$ is lower bounded by some finite constant $\underline{f}$.

The assumption on the lower boundedness of $\Phi(x)$ is equivalent to assume that $h(x)$ is lower bounded for any $x \in \mathbb{R}^{d_x}$ satisfying $c_j(x) \le 0, \forall j \in \mathcal{J}$.

*Remark* 1. In our problem (2), while $\mathcal{Y}$ serves as the feasible set of Lagrange multipliers that are inherently unbounded, the boundedness of the optimal dual set has been justified in Nedić & Ozdaglar (2009) under the Slater condition. Therefore, this assumption is commonly imposed in existing work (Liu et al., 2020b; Xu et al., 2023), and we can construct a bounded set containing the optimal dual variables to replace $\{y \in \mathbb{R}^{d_y} | y \ge 0\}$ in our method.

**Assumption 2.2.** $f(x,y)$ is differentiable and Lipschitz continuous, i.e., for any $(x,y) \in \mathbb{R}^{d_x} \times \mathcal{Y}$, we have $\|\nabla_x f(x,y)\| \le \Lambda$ and $\|\nabla_y f(x,y)\| \le \Lambda$ for some $\Lambda > 0$.

**Assumption 2.3.** $f(x,y)$ is $L$-smooth in $x$ and $y$, i.e., there exist some $L \ge 0$ satisfying $\|\nabla f(x_1, y_1) - \nabla f(x_2, y_2)\| \le L(\|x_1 - x_2\| + \|y_1 - y_2\|)$ for any $x_1, x_2 \in \mathbb{R}^{d_x}$, and $y_1, y_2 \in \mathcal{Y}$.

Assumptions 2.2-2.3 impose the Lipschitz continuity on $f(x,y)$ and its gradients, which are standard in the literature of both first-order and zeroth-order optimization (Nedić & Ozdaglar, 2009; Ghadimi & Lan, 2013; Zhou et al., 2025). Similarly, we can also equivalently impose Lipschitz continuity on $h(x)$, $c_j(x), \forall j \in \mathcal{J}$ and their gradients to replace Assumptions 2.2 and 2.3.

For min-max problems, a widely adopted stationarity measure is the proximal gradient for first-order and zeroth-order nonconvex optimization (Lin et al., 2020; Liu et al., 2020b; Xu et al., 2023):

$$\mathfrak{g}(x,y) = \begin{pmatrix} \mathfrak{g}_x(x,y) \\ \mathfrak{g}_y(x,y) \end{pmatrix} = \begin{pmatrix} \nabla_x f(x,y) \\ (1/\beta)(y - \mathcal{P}_{\mathcal{Y}}[y + \beta \nabla_y f(x,y)]) \end{pmatrix},$$

where $\beta$ is the step size for dual updates. A point $(x,y) \in \mathbb{R}^{d_x} \times \mathcal{Y}$ is a first-order stationary point of (2) if $\|\mathfrak{g}(x,y)\| = 0$. We also introduce another notion of stationarity measure. The problem (2) is equivalent to minimizing the function $\Phi(x) = \max_{y \in \mathcal{Y}} f(x,y)$ over $\mathbb{R}^{d_x}$. The norm of $\nabla \Phi(x)$ is an appropriate stationarity measure for nonconvex optimization when $\Phi(x)$ is differentiable (Wang et al., 2023). However, $\Phi(x)$ may fail to be differentiable even if $f(x,y)$ is concave in $y$, as the maximum may not be uniquely attained. Alternatively, we define the Moreau envelope of $\Phi(x)$ for any $\lambda > 0$ as

$$\Phi_\lambda(x) = \min_{u \in \mathbb{R}^{d_x}} \left\{ \Phi(u) + \frac{1}{2\lambda} \|u - x\|^2 \right\}.$$

The Moreau envelope $\Phi_{1/2L}(x)$ with parameter $\frac{1}{2L}$ and $\nabla\Phi_{1/2L}(x)$ are both well-defined because $\Phi(u) + L\|u - x\|^2$ is strongly convex in $u$ given $x$. Furthermore, $\Phi_{1/2L}(x)$ is differentiable and smooth in $x$. A point $x \in \mathbb{R}^{d_x}$ is a stationary point of $\Phi$ if $\|\nabla\Phi_{1/2L}(x)\| = 0$. This stationarity measure is also widely used in nonconvex-concave settings (Mahdavinia et al., 2022; Davis & Drusvyatskiy, 2019). As shown in Lin et al. (2020), the computational overhead of transferring this notion to the one measured by $\|\mathfrak{g}(x, y)\|$ is negligible compared to the overall query complexity. Therefore, we define our stationarity measure as follows.

**Definition 2.1.** Let $M(x, y) = \min\{\|\mathfrak{g}(x, y)\|, \|\nabla\Phi_{1/2L}(x)\|\}$ for some $(x, y) \in \mathbb{R}^{d_x} \times \mathcal{Y}$. We say a point $(x, y)$ is an $\epsilon$-stationary point of (2) if $M(x, y) \leq \epsilon$.

## 2.2 Zeroth-Order Gradient Estimation

Various gradient estimators for ZO have been proposed in the literature, where the two-point RGE is most widely applied. For a differentiable function $h : \mathbb{R}^{d_x} \to \mathbb{R}$, the two-point RGE is defined as

$$g(x; r, z) = \frac{h(x + rz) - h(x)}{r} \cdot z, \tag{3}$$

where $g(x; r, z) \in \mathbb{R}^{d_x}$ is the estimated gradient for $\nabla h(x)$. Here, $r > 0$ is the smoothing radius and $z \in \mathbb{R}^{d_x}$ is a random perturbation vector typically sampled from the Gaussian distribution $\mathcal{N}(0, I_{d_x})$ or the uniform distribution on a sphere with radius $\sqrt{d_x}$ (Nesterov & Spokoiny, 2017; Duchi et al., 2015). When $h$ is smooth, this estimator enjoys a bias bounded by the smoothing radius (Malik et al., 2020). In contrast, CGE adds perturbation to each dimension separately and applies the finite difference of function values to construct a full gradient. The CGE is defined as

$$g(x; r, \{e_i\}_{i=1}^{d_x}) = \sum_{i \in [d_x]} \frac{h(x + re_i) - h(x)}{r} \cdot e_i, \tag{4}$$

where $g(x; r, \{e_i\}_{i=1}^{d_x}) \in \mathbb{R}^{d_x}$ approximates $\nabla h(x)$, and $e_i \in \mathbb{R}^{d_x}$ is the unit vector with only the $i$th entry being 1. Let $g_i(x; r, e_i)$ denote the $i$th entry of $g(x; r, \{e_i\}_{i=1}^{d_x})$. Similarly, the bias of CGE is also negligible given a small smoothing radius (Berahas et al., 2022).

**Dilemma of Trading off Single-Step and Overall Query Complexities.** Both RGE and CGE have biases bounded by the smoothing radius $r$. RGE in (3) is efficient for a single step and only requires two function values to construct a gradient. Its variance approximately takes the form $\mathcal{O}(d)\|\nabla h(x)\|^2$ (Liu et al., 2020a). In unconstrained problems, we have $\|\nabla h(x^*)\| = 0$ for any optimal solution $x^*$. Therefore, the variance is negligible as the iterates approach the optimal solution, which allows RGE-based algorithms to mimic their first-order counterparts and achieve similar convergence results. However, in constrained problems, the above property does not hold, as the gradient $\nabla h(x^*)$ may not be zero. The large variance of RGE leads to worse overall query complexities in constrained problems (Nguyen & Balasubramanian, 2023). In contrast, the variance of CGE is controlled by the order of $\mathcal{O}(r^2)$ and is negligible with small $r$. Therefore, CGE-based algorithms generally enjoy the state-of-the-art overall query complexity bounds (Xu et al., 2024; Zhou et al., 2025). However, CGE requires $\mathcal{O}(d)$ function values to construct a full gradient, which is inefficient for large $d$. As a result, achieving efficiency in both aspects has yet to be addressed.

# 3 Zeroth-Order Block Gradient Descent Ascent

In this section, we leverage BCGEs and block updates to design a new algorithm with controllable single-step efficiency. Then, we establish its convergence guarantee and query complexity bound.

## 3.1 Algorithm Design

We propose the *zeroth-order block gradient descent ascent* (ZOB-GDA) algorithm as presented in Algorithm 1 for solving (2). Our algorithm follows the main steps of standard GDA, which perform the gradient descent in $x$ and gradient ascent in $y$. However, ZOB-GDA differs from the conventional zeroth-order GDA method introduced in Liu et al. (2020b), which employs the RGE in (3). Instead

of estimating a full gradient, ZOB-GDA randomly selects a block of dimensions to estimate the partial gradients and performs block descent ascent at each step. Specifically, for the $k$th iterate $(x_k, y_k)$, we randomly sample a block of dimensions, denoted by a set $\mathcal{I}_k \subseteq [d_x]$ with $|\mathcal{I}_k| = b$ ($b$ is an integer ranging from 1 to $d_x$). Then, we apply BCGE to estimate the block gradient of $f(x_k, y_k)$:

$$G_x^{\mathcal{I}_k}(x_k, y_k) = \sum_{i \in \mathcal{I}_k} \frac{f(x_k + r_k e_i, y_k) - f(x_k, y_k)}{r_k} e_i,$$

where we denote $G_x^{\mathcal{I}_k}(x_k, y_k) \in \mathbb{R}^{d_x}$ as the vector with the entries of $\mathcal{I}_k$ being estimated and other entries being 0. We can apply different smoothing radii $r_k$ for each iteration. For simplicity, we denote $G_x(x_k, y_k) = G_x^{\mathcal{I}}(x_k, y_k)$ when $\mathcal{I} = [d_x]$. For the update of the dual variable $y$, we have the partial gradient $\nabla_y f(x_k, y_k) = c(x_k)$. No additional queries are required as $c(x_k)$ has been observed when computing $G_x^{\mathcal{I}_k}(x_k, y_k)$.

Specifically, when the block size satisfies $b = 1$, the primal update resembles a coordinate update. When $b = d_x$, the primal update uses a full gradient and resembles the primal update in traditional CGE-based algorithms (Xu et al., 2024; Zhou et al., 2025). Moreover, we can control the single-step efficiency by adjusting the block size $b$. It will be shown that the choice of $b$ does not affect the overall query complexity of ZOB-GDA.

Unlike existing ZO literature that applies coordinate/block updates, we first generalize them within the GDA framework to address non-analytical constraints, whose dynamics and convergence analysis are significantly more complicated due to the coupling of primal and dual steps.

---

**Algorithm 1** Zeroth-order block gradient descent ascent (ZOB-GDA)

---

1: **Input:** Initial $(x_0, y_0) \in \mathbb{R}^{d_x} \times \mathcal{Y}$, maximum steps $K$, block size $b$, and the step sizes $\alpha, \beta$.
2: **for** $k = 0, 1, 2, \cdots, K - 1$ **do**
3:     Randomly sample $\mathcal{I}_k \subseteq [d_x]$ with $|\mathcal{I}_k| = b$ and update $x_k$ by

$$x_{k+1} = x_k - \alpha \cdot G_x^{\mathcal{I}_k}(x_k, y_k). \tag{5}$$

4:     Update $y_k$ by $y_{k+1} = \mathcal{P}_{\mathcal{Y}}\left[y_k + \beta \cdot \nabla_y f(x_k, y_k)\right]$.
5: **end for**
6: **Output:** $\{(x_k, y_k)\}_{k=0}^{K}$

---

### 3.2 Convergence Results of ZOB-GDA

In this subsection, we establish convergence guarantees and query complexity bounds for Algorithm 1 in nonconvex-concave cases. First, we provide the following lemma to bound the bias of coordinate gradient estimation.

**Lemma 3.1.** *For a $L$-smooth and differentiable function $h : \mathbb{R}^{d_x} \to \mathbb{R}$, i.e., $\|\nabla h(x) - \nabla h(x')\| \leq L\|x - x'\|, \forall x, x' \in \mathbb{R}^{d_x}$, we have $|\nabla_i h(x) - g_i(x; r, e_i)| \leq \frac{1}{2}Lr$.*

Lemma 3.1 and its extended versions have appeared in existing literature (Lian et al., 2016; Berahas et al., 2022; Jin et al., 2023), thus, we omit its proof here. Lemma 3.1 demonstrates that $g_i(x; r, e_i)$ is a good partial gradient estimator, in the sense that both its bias and variance can be effectively controlled by the smoothing radius and $L$. This error bound plays a fundamental role and will be frequently used in our theoretical analysis.

Let $N = \frac{d_x}{b}$, $R_y = \|\bar{y}\|$ and $\Lambda_0 = \Lambda + \frac{\sqrt{b}L}{2} \cdot \sup_k\{r_k\}$. Then, we present the following Theorem to characterize the convergence of ZOB-GDA. Its proof is provided in Appendix B.

**Theorem 3.1.** *Suppose Assumptions 2.1-2.3 hold. The sequence $\{(x_k, y_k)\}_{k=0}^{K}$ is generated by ZOB-GDA. The step sizes satisfy $0 < \alpha, \beta \leq 1/L$, and the sequence of smoothing radii satisfies $\sum_{k=0}^{K} r_k^2 < \infty$. Then, we have*

$$\min_{k \leq K-1} \mathbb{E}\left[\|\Phi_{1/2L}(x_k)\|\right] \leq \mathcal{O}\left(\sqrt{\frac{N}{\alpha K}}\right) + \epsilon_c,$$

*where $\epsilon_c = \left(16L\Lambda_0 R_y \sqrt{2\alpha/\beta} + 48\alpha L\Lambda_0^2\right)^{1/2}$.*

The results in Theorem 3.1 imply that ZOB-GDA can converge within a fixed error at a convergence rate of $\mathcal{O}(\sqrt{N/\alpha K})$. The query complexity required to achieve $\min_{k \leq K-1} \mathbb{E}\left[M(x_k, y_k)\right] \leq \epsilon + \epsilon_c$ is $\mathcal{O}(d/\epsilon^2)$. The fixed error results from the use of constant step sizes. Given the expression of $\epsilon_c$, one could adopt diminishing or small step sizes for $\alpha$ to eliminate the term $\epsilon_c$. The following corollary characterizes the exact convergence guarantee of ZOB-GDA.

**Corollary 3.2.** *Suppose that the conditions in Theorem 3.1 hold. Further set the step size* $\alpha = \mathcal{O}\left((N/K)^{\frac{2}{3}}\right)$. *Then, we have* $\min_{k \leq K-1} \mathbb{E}\left[\left\|\nabla \Phi_{1/2L}(x_k)\right\|\right] \leq \mathcal{O}\left((N/K)^{\frac{1}{6}}\right)$.

The derivation of Corollary 3.2 is straightforward by substituting the step size into the result in Theorem 3.1. The result in Corollary 3.2 shows that using two-time-scale step sizes for the updates of $x$ and $y$ can effectively eliminate the fixed error term $\epsilon_c$. This phenomenon aligns with the first-order GDA in nonconvex-concave settings (Lin et al., 2020). Building upon Corollary 3.2, we obtain the following corollary to establish the overall query complexity of ZOB-GDA.

**Corollary 3.3.** *Suppose the conditions in Theorem 3.1 hold. Set* $\alpha = \mathcal{O}\left(\epsilon^4\right)$ *for any sufficiently small* $\epsilon$. *Then, the query complexity to achieve* $\min_{k \leq K-1} \mathbb{E}\left[\left\|\nabla \Phi_{1/2L}(x_k)\right\|\right] \leq \epsilon$ *is* $\mathcal{O}\left(\frac{d_x}{\epsilon^6}\right)$.

To the best of our knowledge, Corollary 3.3 establishes the first query complexity result for zeroth-order algorithms in the standard GDA framework for nonconvex-concave settings. Notably, this complexity differs from the first-order GDA by an additional factor $d_x$ (Lin et al., 2020), which is inherent to zeroth-order gradient estimation. Compared to the ZOAGP algorithm (Xu et al., 2024) with the query complexity bound $\mathcal{O}(d/\epsilon^4)$, ZOB-GDA's complexity bound seems worse due to the limitation of standard GDA framework. However, the single-step gradient estimation can be significantly more efficient for ZOB-GDA by using a small $b$. In the next section, we will leverage block updates with a variant of the GDA framework to design a new algorithm that achieves both the best-known overall query complexity and adjustable single-step efficiency.

## 4 ZEROTH-ORDER BLOCK SMOOTHED GRADIENT DESCENT ASCENT

In this section, we leverage block updates with a variant of GDA, smoothed GDA, to design a new algorithm, and show the best-known convergence result for solving problem (1).

### 4.1 ALGORITHM DESIGN

Before presenting our algorithm, we define the smoothed function of $f(x, y)$ as

$$K(x, y; z) = f(x, y) + \frac{p}{2}\|x - z\|^2,$$

for some auxiliary variable $z \in \mathbb{R}^{d_x}$. The squared term can introduce strong convexity and further smoothness in $x$ with a proper $p$. Then, we will perform gradient descent ascent on the smoothed function $K(x, y; z)$, which is inspired by the first-order smoothed GDA (SGDA) in Zhang et al. (2020). The *zeroth-order block smoothed gradient descent ascent* (ZOB-SGDA) algorithm is proposed as shown in Algorithm 2. Similarly to ZOB-GDA, we randomly sample a block of dimensions $\mathcal{I}_k$ for primal variables and only update the selected dimensions using BCGEs at each step. We denote the partial gradients along the dimensions $\mathcal{I}_k$ as

$$G_x^{\mathcal{I}_k}(x_k, y_k; z_k) = \sum_{i \in \mathcal{I}_k}\left(\frac{f(x_k + r_k e_i, y_k) - f(x_k, y_k)}{r_k}e_i + p(x_k - z_k) \odot e_i\right),$$

where $\odot$ denotes the Hadamard (element-wise) product. We also denote $G_x(x_k, y_k; z_k) = G_x^{\mathcal{I}}(x_k, y_k; z_k)$ when $\mathcal{I} = [d_x]$. The update of $y_k$ follows the same way in ZOB-GDA. Additionally, an extra update for $z_k$ is introduced by an averaging step. When $\gamma = 1$, it is obvious that ZOB-SGDA can resemble ZOB-GDA.

### 4.2 CONVERGENCE ANALYSIS OF ZOB-SGDA

By properly setting the parameters for ZOB-SGDA, we can establish its convergence result as summarized in Theorem 4.1. Its proof is provided in Appendix C.

---

**Algorithm 2** Zeroth-Order Block Smoothed Gradient Descent Ascent (ZOB-SGDA)

---

1: **Input:** Initial $(x_0, y_0) \in \mathbb{R}^{d_x} \times \mathcal{Y}$, $z_0 = x_0$, maximum steps $K$, block size $b$, and the step sizes $\alpha, \beta, \gamma$.
2: **for** $k = 0, 1, 2, \cdots, K - 1$ **do**
3:      Randomly sample a set $\mathcal{I}_k \subseteq [d_x]$ with $|\mathcal{I}_k| = b$ and update $x_k$ by

$$x_{k+1} = x_k - \alpha \cdot G_x^{\mathcal{I}_k}(x_k, y_k; z_k). \tag{6}$$

4:      Update $y_k$ by $y_{k+1} = \mathcal{P}_{\mathcal{Y}}[y_k + \beta \cdot \nabla_y K(x_k, y_k; z_k)]$.
5:      Update $z_k$ by $z_{k+1} = \gamma x_{k+1} + (1 - \gamma) z_k$.
6: **end for**
7: **Output:** $\{(x_k, y_k)\}_{k=0}^{K}$

---

**Theorem 4.1.** *Suppose Assumptions 2.1 and 2.3 hold. The sequence $\{(x_k, y_k)\}_{k=0}^{K}$ is derived from ZOB-SGDA. Set the parameters $p \geq 3L$, $\sum_{k=0}^{K} r_k^2 \leq \frac{1}{b}$, and $\alpha \leq \frac{1}{p+10L+1}$. Furthermore, let $\beta \leq \min\left\{\frac{1}{12L}, \frac{\alpha^2(p-L)^2}{4L(\sqrt{N}+\alpha(p-L))^2}\right\}$, and $\gamma \leq \min\left\{\sqrt{\frac{1}{KN}}, \frac{1}{36}, \frac{1}{768p\beta}\right\}$. Then, we have*

$$\min_{k \leq K-1} \mathbb{E}\left[\|\mathfrak{g}(x_k, y_k)\|\right] \leq \mathcal{O}\left(\left(\frac{N}{K}\right)^{1/4}\right).$$

The results in Theorem 4.1 show that ZOB-SGDA can converge to a stationary point at the convergence of $\mathcal{O}((N/K)^{\frac{1}{4}})$. Note that we do not impose the Lipschitz assumption on $f(x, y)$ in Theorem 4.1. Similarly, based on Theorem 4.1, we have the following corollary to characterize the query complexity of ZOB-SGDA.

**Corollary 4.2.** *Suppose that the conditions in Theorem 4.1 hold. For any sufficiently small $\epsilon > 0$, set $\alpha, \beta, r_k$ as in Theorem 4.1 and $\gamma = \mathcal{O}(\epsilon^2/N)$. Then, the query complexity to achieve $\min_{k \leq K-1} \mathbb{E}\left[\|\mathfrak{g}(x_k, y_k)\|\right] \leq \epsilon$ is $\mathcal{O}\left(\frac{d_x}{\epsilon^4}\right)$.*

We can see from Corollary 4.2 that ZOB-SGDA has the query complexity bound $\mathcal{O}(d_x/\epsilon^4)$, regardless of the choice of block sizes. That means our algorithm can achieve the best-known overall query complexity while maintaining controllable single-step efficiency. For instance, only two queries are required for each step when we set $b = 1$, which is more efficient than other CGE-based algorithms that require $\mathcal{O}(d)$ queries for a gradient estimation.

## 5 DISCUSSIONS

**Stationary Points of (2) Can Provide Solutions to (1).** Our theoretical results establish convergence guarantees to stationary points of $f(x, y)$ for the proposed algorithms, while the convergence guarantees to the solutions to problem (1) are yet to be established. However, under proper conditions, the stationary points of $f(x, y)$ satisfying $\|\mathfrak{g}(x, y)\| = 0$ is also a critical KKT point of problem (1). We provide the following lemma to characterize this property. The definition of critical KKT points and the proof of Lemma 5.1 are provided in Appendix D.

**Lemma 5.1.** *Suppose that $(x, y) \in \mathbb{R}^{d_x} \times \mathcal{Y}$ is a stationery point of $f(x, y)$ satisfying $\|\mathfrak{g}(x, y)\| = 0$ and $y < \overline{y}$. Then, $x$ is a critical KKT point of problem (1).*

The condition $y < \overline{y}$ stems from the gap between our assumption that $\mathcal{Y}$ is bounded and the fact that the multiplier $y \geq 0$ is generally not in practice. This is a common and fundamental gap in the analysis of GDA-type algorithms (Nedić & Ozdaglar, 2009; Liu et al., 2020b; Xu et al., 2024), which we believe is an important and interesting future research direction.

**Extend Block Updates to Broader Problem Settings.** In our problem (1), we deal with all constraints in the general form $c_j(x) \leq 0$. If equality constraints $c_j(x) = 0$ have to be considered, we can incorporate them by adding two inequalities $c_j(x) \geq 0$ and $c_j(x) \leq 0$. Besides, we can also consider some simple constraints by constraining the feasible space directly:

$$\min_{x \in \mathcal{X}} h(x) \qquad \text{s.t. } c_j(x) \leq 0, \ \forall j \in \mathcal{J},$$

and deal with $x \in \mathcal{X}$ by projection, i.e., $x_{k+1} = \mathcal{P}_{\mathcal{X}}[x_k - \alpha G_x^{\mathcal{I}_k}(x_k, y_k)]$. We can get the same theoretical results in our analysis when $\mathcal{X}$ is *convex* and *decomposable*, i.e., $\mathcal{X} = \prod_{i \in [d_x]} \mathcal{X}_i$. This requirement originates from the fundamental limit of block/coordinate updates (Lian et al., 2016; Jin et al., 2023). Note that we need to make the modifications: $\mathfrak{g}_x(x, y) = \frac{1}{\alpha}(x - \mathcal{P}_{\mathcal{X}}[x - \alpha \nabla_x f(x, y)])$, and $\Phi_{1/2L}(x) = \min_{u \in \mathcal{X}} \left\{ \Phi(u) + \frac{1}{2\lambda}\|u - x\|^2 \right\}$ for the stationarity measure. The extended theoretical results are straightforward to establish based on our analysis and the non-expansiveness of projection operators; thus, we omit the detailed analysis in this study.

Our algorithms can also be applied to stochastic cases, i.e., $h(x) = \mathbb{E}[h(x; \xi)], c_j(x) = \mathbb{E}[c_j(x; \xi)]$, where $\xi$ is a random variable defined in a probability space. However, we cannot expect better convergence guarantees than the RGE-based algorithms, because extra variance arises in BCGEs due to the stochasticity of $\xi$, which diminishes the advantage of controllable variance of our methods.

# 6 NUMERICAL SIMULATIONS

We validate our algorithms through numerical experiments on an energy management problem in a 141-bus distribution network with $d_x = 168$ (Khodr et al., 2008; Zhou et al., 2025). In this problem, the goal is to adjust the load of multiple users within a distribution network to curtail a specific amount of load while minimizing the cost of participating in load curtailment. The detailed problem formulation and experimental settings are provided in Appendix E.

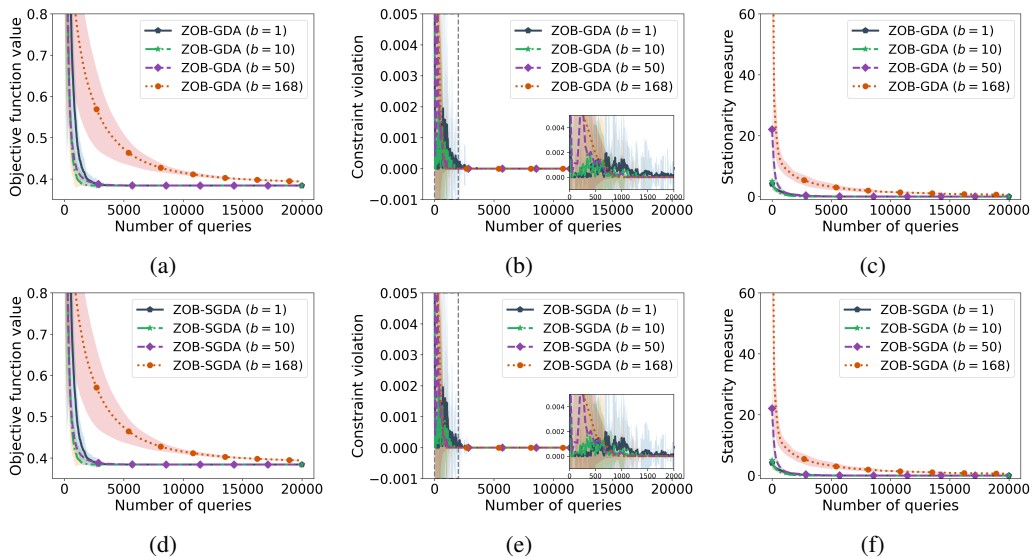

Figure 1: Performance of ZOB-GDA and ZOB-SGDA. (a), (b), and (c) present the objective function value, constraint violation, and stationarity measure of ZOB-GDA. (d), (e), and (f) present the corresponding results for ZOB-SGDA.

First, we apply ZOB-GDA and ZOB-SGDA to solve the problem using different block sizes. Their performance is averaged over 50 repeated runs with different initial parameters and shown in Figure 1. The dark curves represent the average performance, and the shaded areas represent the standard deviation. The results show that both ZOB-GDA and ZOB-SGDA can converge to the same objective function value with the constraint satisfied. Their stationarity measures can both converge to 0, which validates our theoretical guarantees. While different block sizes lead to convergence to the same objective, properly selecting the block sizes may improve the query complexity.

We also compare our algorithms (block size $b = 10$) with three others, i.e., ZO-MinMax (Liu et al., 2020b), SZO-ConEx (Nguyen & Balasubramanian, 2023), and ZOAGP (Xu et al., 2024), which can be applied to solve problem (1). Each algorithm is tested with 50 runs, and the average performance is presented in Figure 2. The results show that our algorithms can both converge to a solution significantly faster than other algorithms. To better compare their query complexities, we present the average number of queries required to generate solutions with different qualities, as in Table 2. It

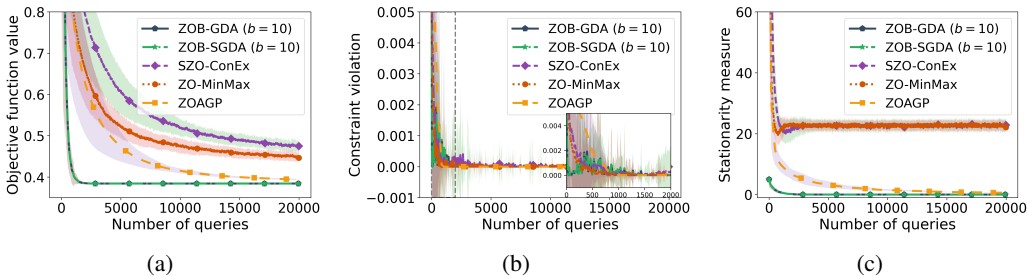

Figure 2: Performance comparison of different algorithms.

is shown that ZOB-GDA and ZOB-SGDA exhibit highly similar performance under different block sizes, while the complexity bound of the latter is theoretically tighter. Notably, even though different block sizes share the same query complexity bound, proper block sizes may lead to much improved performance (over 10 times better than existing methods). Moreover, SZO-ConEx and ZOMinMax have the worst performance due to the large variance of RGEs.

In practice, observations from real systems are often noisy, which can influence the accuracy of gradient estimations. Our algorithms can still exhibit robust and satisfactory performance under noisy cases. Due to space limitations, the test results are provided in Appendix E.3.

Table 2: Average numbers of iterations and queries required to generate solutions with certain levels of relative errors and zero constraint violation. "NaN" means no runs can achieve such a solution.

| Relative error | | 10% | | 1% | | 0.1% | |
|---|---|---|---|---|---|---|---|
| | | Iteration | Complexity | Iteration | Complexity | Iteration | Complexity |
| ZOBGDA | $b = 1$ | 722.06 | 1444.12 | 1213.48 | 2426.96 | 1584.84 | 3169.68 |
| | $b = 10$ | 73.66 | 810.26 | 130.70 | 1437.70 | 163.78 | 1801.58 |
| | $b = 50$ | 21.92 | 1117.92 | 57.24 | 2919.24 | 76.16 | 3884.16 |
| | $b = 168$ | 51.48 | 8700.12 | 185.92 | 31420.48 | 265.71 | 44905.71 |
| ZOB-SGDA | $b = 1$ | 722.80 | 1445.60 | 1195.94 | 2391.88 | 1594.32 | 3188.64 |
| | $b = 10$ | 74.00 | 814.00 | 131.90 | 1450.90 | 169.68 | 1866.48 |
| | $b = 50$ | 22.18 | 1131.18 | 58.26 | 2971.26 | 77.76 | 3965.76 |
| | $b = 168$ | 52.06 | 8798.14 | 187.42 | 31673.98 | 266.59 | 45054.15 |
| ZO-MinMax | | 12535.10 | 25070.20 | 12771.33 | 25542.65 | NaN | NaN |
| SZO-ConEx | | 12817.62 | 51270.49 | NaN | NaN | NaN | NaN |
| ZOAGP | | 51.48 | 8700.12 | 185.92 | 31420.48 | 265.71 | 44905.71 |

## 7 CONCLUSION

In this research, we study a general optimization problem with black-box constraints. We reformulate it as a min-max problem, and then apply zeroth-order optimization (ZO) methods to solve it using the input-output information. Specifically, by integrating block updates with gradient descent ascent (GDA), we develop two novel algorithms, called ZOB-GDA and ZOB-SGDA, which achieve efficiency in both single-step gradient estimation and the overall query complexity. Our theoretical results demonstrate that ZOB-GDA achieves the same query complexity as its first-order counterpart with an additional dimension-dependent factor, and ZOB-SGDA enjoys the best-known complexity bound. In addition, our numerical experiments validate the superior performance of our algorithms. However, our work on block updates in constrained ZO is just a beginning. There are still several open challenges. First, while the block update framework is a broadly applicable technique for improving single-step efficiency, its integration with other primal–dual algorithms requires more study. Second, although we anticipate that the benefits of block updates in stochastic constrained ZO will be more limited than in deterministic settings, rigorous validation requires further investigation.

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

## A  RELATED WORK

Here, we provide a detailed discussion on prior related work in zeroth-order optimization, zeroth-order gradient descent ascent, and coordinate/block updates in ZO.

**Zeroth-Order Optimization.**  ZO has emerged as a prevalent tool to solve black-box problems and found application across machine learning (Liu et al., 2020a; Nguyen & Balasubramanian, 2023), power systems (Hu et al., 2024; Zhou et al., 2025), simulation optimization (Fu et al., 2015; Lam & Zhang, 2024), large language models (Malladi et al., 2023; Zhang et al., 2024), etc. ZO originates from the stochastic approximation method in Kiefer & Wolfowitz (1952), where CGE is applied to estimate partial gradients along all dimensions via finite differences of function values. This is inefficient for high-dimensional problems, even if parallel techniques can be applied (Scheinberg, 2022). To improve single-step efficiency, one-point and two-point RGEs have been developed by estimating gradients along randomized directions (Flaxman et al., 2005; Nesterov & Stich, 2017; Lam & Zhang, 2024). Generally, RGE-based algorithms can achieve the same oracle complexities as their first-order counterparts in unconstrained problems, differing by a dimension-dependent factor (Liu et al., 2020a). However, RGEs suffer from large variance in gradient estimation in constrained problems (see Section 2.2 for a detailed discussion). Moreover, most literature considers simple constraints on the input $x$ that can be dealt with by projection operations (Duchi et al., 2015; Yuan et al., 2015; Jin et al., 2023; He et al., 2024), which cannot be used to solve problem (1).

**Zeroth-Order GDA.**  GDA is a classical framework for solving min-max problems and has been extensively studied (Nemirovski, 2004; Nedić & Ozdaglar, 2009; Lin et al., 2020; Xu et al., 2023; Zhang et al., 2020). It is also well-established and widely applied to solve zeroth-order min-max problems of the form (2) (Hu et al., 2024; Nguyen & Balasubramanian, 2023). The authors of Liu et al. (2018) applied the two-point RGE to solve a composite optimization problem. Then, the standard zeroth-order GDA was applied to the general min-max problems (Liu et al., 2020b; Wang et al., 2023), while only the query complexity of nonconvex-strongly concave cases was established. Several variants of zeroth-order GDA have been developed for convex-concave settings, such as zeroth-order OGDA-RR (Maheshwari et al., 2022) and zeroth-order extra-gradient (Zhou et al., 2025), which can achieve the query complexity bounds of $\mathcal{O}(d^4/\epsilon^2)$ and $\mathcal{O}(d/\epsilon^2)$ to reach an $\epsilon$-optimal solution, respectively. For nonconvex–concave problems, Xu et al. (2024) proposed combining alternating gradient projection with CGEs to solve a min-max problem with the query complexity bound of $\mathcal{O}(d/\epsilon^4)$ to obtain an $\epsilon$-stationary point. In Nguyen & Balasubramanian (2023), the SZO-ConEX algorithm was designed based on RGEs to achieve the query complexity of $\mathcal{O}(d/\epsilon^6)$ to derive an $\epsilon$-critical Karush-Kuhn-Tucker (KKT) point of the problem (1).

**Zeroth-Order Coordinate/Block Updates.**  The framework of coordinate/block updates is widely adopted in first-order optimization (Nesterov & Stich, 2017; Latafat et al., 2019). The core idea is to apply the partial gradients along a subset of full dimensions to update the iterates. The applications of coordinate/block updates in ZO mainly lie in unconstrained problems (Lian et al., 2016; Cai et al., 2021), where only the coordinate/block gradients along a subset are estimated using block CGEs or RGEs at each step. Their extension to constrained problems, however, remains relatively underexplored. In Shanbhag & Yousefian (2021), the RGE was combined with zeroth-order block updates and projected gradient descent to solve a stochastic constrained problem. Moreover, in He et al. (2024) and Jin et al. (2023), a cyclic zeroth-order block coordinate descent method and a randomized zeroth-order coordinate descent method were proposed, respectively, to solve the deterministic constrained problems and achieve complexity bounds proportional to $\epsilon^{-2}$ for nonconvex optimization. However, all these methods require the constraint set to be projection-friendly and coordinate/block-structured, which is usually too restrictive in practical problems and inapplicable to non-analytical constraint sets (such as in our problem (1)).

## B  PROOF OF THEOREM 3.1

First, define the proximity operator

$$\text{prox}_{\lambda h}(x) = \arg\min_{u \in \mathbb{R}^{d_x}} \left\{ h(u) + \frac{1}{2\lambda} \|x - u\|^2 \right\}.$$

Define the filtration: $\mathcal{F}_k = \sigma(x_0, y_0, \mathcal{I}_0, \cdots, i_{k-1}, x_k, y_k)$. Then, we provide the following lemma to bound the one-step drift of $\Phi_{1/2L}(x_k)$.

**Lemma B.1.** *Let $\Delta_k = \Phi(x_k) - f(x_k, y_k)$. The following inequality holds for any $k \geq 0$,*

$$\mathbb{E}\left[\Phi_{1/2L}(x_{k+1}) - \Phi_{1/2L}(x_k)\big|\mathcal{F}_k\right]$$

$$\leq \frac{2\alpha L}{N}\Delta_k - \frac{\alpha}{8N}\left\|\nabla\Phi_{1/2L}(x_k)\right\|^2 + \frac{2\alpha^2\Lambda^2 L}{N} + \frac{\alpha^2 bL^3 r_k^2}{2} + \frac{\alpha bL^2 r_k^2}{2}. \tag{7}$$

The proof of Lemma B.1 is delayed in Appendix B.1. We further provide the following lemma to bound the summation of $\Delta_k$.

**Lemma B.2.** *For any integer $B$ that can divide $K$, we have*

$$\frac{1}{K}\sum_{k=0}^{K-1}\Delta_k \leq \alpha\Lambda_0^2(B+2) + \frac{R_y^2}{2\beta B} + \frac{\Delta_0}{K}.$$

The proof of Lemma B.2 is provided in Appendix B.2. Taking the telescoping sum of (7) and taking the total expectation, we have

$$\frac{1}{K}\sum_{k=0}^{K-1}\mathbb{E}\left[\left\|\nabla\Phi_{1/2L}(x_k)\right\|^2\right]$$

$$\leq \frac{8N}{\alpha K}\mathbb{E}\left[\Phi_{1/2L}(x_0) - \Phi_{1/2L}(x_K)\right] + \frac{16L}{K}\sum_{k=0}^{K-1}\mathbb{E}\left[\Delta_k\right]$$

$$+ 16\alpha\Lambda^2 L + \frac{4\alpha d_x L^3}{K}\sum_{k=1}^{K-1}r_k^2 + \frac{4d_x L^2}{K}\sum_{k=0}^{K-1}r_k^2$$

$$\leq \frac{8N\Delta_\Phi}{\alpha K} + \frac{16L}{K}\sum_{k=0}^{K-1}\mathbb{E}\left[\Delta_k\right] + 16\alpha\Lambda^2 L + \frac{8d_x L^2}{K}\sum_{k=0}^{K-1}r_k^2,$$

where $\Delta_\Phi := \Phi_{1/2L}(x_0) - \min_{x\in\mathbb{R}^{d_x}}\Phi_{1/2L}(x)$. The last step follows from the fact that $\alpha L \leq 1$. Then, we can combine the above inequality with the result in Lemma B.2 to get

$$\frac{1}{K}\sum_{k=0}^{K-1}\mathbb{E}\left[\left\|\nabla\Phi_{1/2L}(x_k)\right\|^2\right]$$

$$\leq \frac{8N\Delta_\Phi}{\alpha K} + \frac{16L\Delta_0}{K} + \frac{8NL^2}{K} + \frac{8LR_y^2}{\beta B} + 16\alpha L\Lambda_0^2(B+3), \tag{8}$$

Due to that, for any $x \in \mathbb{R}^{d_x}$,

$$\Phi_{1/2L}(x) = \min_{u\in\mathbb{R}^{d_x}}\left\{\Phi(u) + L\|x-u\|^2\right\}$$

$$= \min_{u\in\mathbb{R}^{d_x}}\left\{\max_{y\in\mathcal{Y}} f(u,y) + L\|x-u\|^2\right\}$$

$$\geq \min_{u\in\mathbb{R}^{d_x}}\left\{\underline{f} + L\|x-u\|^2\right\}$$

$$= \underline{f},$$

we have $\Delta_\Phi$ is upper bounded. Without loss of generality, set $B = \frac{R_y}{\Lambda_0}\sqrt{\frac{1}{2\alpha\beta}}$ that can divide $K$, then we can derive

$$\frac{1}{K}\sum_{k=0}^{K-1}\mathbb{E}\left[\left\|\nabla\Phi_{1/2L}(x_k)\right\|^2\right] \leq \mathcal{O}\left(\frac{N}{\alpha K}\right) + \epsilon_c^2,$$

which leads to

$$\min_{k\leq K-1}\mathbb{E}\left[\left\|\Phi_{1/2L}(x_k)\right\|\right] \leq \left(\mathcal{O}\left(\frac{N}{\alpha K}\right) + \epsilon_c^2\right)^{1/2} \leq \mathcal{O}\left(\sqrt{\frac{N}{\alpha K}}\right) + \epsilon_c.$$

### B.1 PROOF OF LEMMA B.1

Denote $\hat{x}_k = \text{prox}_{\Phi/2L}(x_k)$. Using the definition of $\Phi_{1/2L}(x_{k+1})$, we have

$$\Phi_{1/2L}(x_{k+1}) \leq \Phi(\hat{x}_k) + L\|\hat{x}_k - x_{k+1}\|^2. \tag{9}$$

Based on the update of $x_k$, we have

$$
\begin{aligned}
&\|\hat{x}_k - x_{k+1}\|^2 \\
&= \left\|\hat{x}_k - x_k + \alpha G_x^{\mathcal{I}_k}(x_k, y_k)\right\|^2 \\
&= \|\hat{x}_k - x_k\|^2 + 2\alpha\langle G_x^{\mathcal{I}_k}(x_k, y_k), \ \hat{x}_k - x_k\rangle + \alpha^2 \left\|G_x^{\mathcal{I}_k}(x_k, y_k)\right\|^2.
\end{aligned} \tag{10}
$$

Substituting the above equation into (9) leads to

$$
\begin{aligned}
&\Phi_{1/2L}(x_{k+1}) \\
&\leq \Phi(\hat{x}_k) + L\|\hat{x}_k - x_k\|^2 + \alpha^2 L \left\|G_x^{\mathcal{I}_k}(x_k, y_k)\right\|^2 + 2\alpha L\langle G_x^{\mathcal{I}_k}(x_k, y_k), \ \hat{x}_k - x_k\rangle \\
&= \Phi_{1/2L}(x_k) + \alpha^2 L \left\|G_x^{\mathcal{I}_k}(x_k, y_k)\right\|^2 + 2\alpha L\langle G_x^{\mathcal{I}_k}(x_k, y_k), \ \hat{x}_k - x_k\rangle.
\end{aligned} \tag{11}
$$

Taking the conditional expectation of the third term on the right-hand side of (11), we have

$$
\begin{aligned}
&\mathbb{E}\left[2\alpha L\langle G_x^{\mathcal{I}_k}(x_k, y_k), \ \hat{x}_k - x_k\rangle \big| \mathcal{F}_k\right] \\
&= \frac{2\alpha L}{N}\langle \nabla_x f(x_k, y_k), \hat{x}_k - x_k\rangle + \frac{2\alpha L}{N}\langle G_x(x_k, y_k) - \nabla_x f(x_k, y_k), \ \hat{x}_k - x_k\rangle \\
&\leq \frac{2\alpha L}{N}(f(\hat{x}_k, y_k) - f(x_k, y_k)) + \frac{\alpha L^2}{N}\|\hat{x}_k - x_k\|^2 \\
&\quad + \frac{2\alpha L}{N}\langle G_x(x_k, y_k) - \nabla_x f(x_k, y_k), \ \hat{x}_k - x_k\rangle \\
&\leq \frac{2\alpha L}{N}(f(\hat{x}_k, y_k) - f(x_k, y_k)) + \frac{3\alpha L^2}{2N}\|\hat{x}_k - x_k\|^2 + \frac{\alpha b L^2 r_k^2}{2},
\end{aligned} \tag{12}
$$

where in the first inequality we used the smoothness of $f(x, y)$, and in the second inequality we used the AM-GM inequality and Lemma 3.1. Using the relation $\Phi(\hat{x}_k) \geq f(\hat{x}_k, y_k)$ and the definition of $\hat{x}_k$, we have

$$f(\hat{x}_k, y_k) - f(x_k, y_k) \leq \Phi(\hat{x}_k) - f(x_k, y_k) \leq \Delta_k - L\|\hat{x}_k - x_k\|^2,$$

where we applied the relation $\Phi(\hat{x}_k) \leq \Phi(x_k) - L\|x_k - \hat{x}_k\|^2$ in the last step. Substituting the above inequality into (12), we further use the relation

$$\|\hat{x}_k - x_k\| = \frac{1}{2L}\|\nabla\Phi_{1/2L}(x_k)\|,$$

which is derived from Davis & Drusvyatskiy (2019), to get

$$
\begin{aligned}
&\mathbb{E}\left[2\alpha L\langle G_x^{\mathcal{I}_k}(x_k, y_k), \ \hat{x}_k - x_k\rangle \big| \mathcal{F}_k\right] \\
&\leq \frac{2\alpha L\Delta_k}{N} - \frac{\alpha}{8N}\|\nabla\Phi_{1/2L}(x_k)\|^2 + \frac{\alpha b L^2 r_k^2}{2}.
\end{aligned} \tag{13}
$$

For the term $\alpha^2 L \left\|G_x^{\mathcal{I}_k}(x_k, y_k)\right\|^2$, we have

$$
\begin{aligned}
&\mathbb{E}\left[\alpha^2 L \left\|G_x^{\mathcal{I}_k}(x_k, y_k)\right\|^2 \big| \mathcal{F}_k\right] \\
&\leq \frac{\alpha^2 L}{N}\left\|G_x(x_k, y_k) - \nabla_x f(x_k, y_k) + \nabla_x f(x_k, y_k)\right\|^2 \\
&\leq \frac{2\alpha^2 L\Lambda^2}{N} + \frac{\alpha^2 b L^3 r_k^2}{2},
\end{aligned} \tag{14}
$$

where we applied the Lipschitz continuity and Lemma 3.1 in the last step.

Taking the expectation of (11) conditioned on $\mathcal{F}_k$ and combining it with (13) and (14) can derive the final result in Lemma B.1.

## B.2 Proof of Lemma B.2

We divide $\{\Delta_k\}_{k=0}^{K-1}$ into $K/B$ blocks: $\{\Delta_k\}_{k=0}^{B-1}, \cdots, \{\Delta_k\}_{jB}^{(j+1)B-1}, \cdots, \{\Delta_k\}_{K-B}^{K-1}$, with each block containing $B$ terms. Then, we have

$$\frac{1}{K} \sum_{k=0}^{K-1} \Delta_k = \frac{B}{K} \sum_{j=0}^{K/B-1} \left( \frac{1}{B} \sum_{k=jB}^{(j+1)B-1} \Delta_k \right). \tag{15}$$

We provide the following lemma to bound $\Delta_k$, whose proof is provided in B.3.

**Lemma B.3.** *Denote $y^*(x)$ as an arbitrary element in the set $\mathcal{Y}^*(x) = \arg\max_{y \in \mathcal{Y}} f(x, y)$ for any $x \in \mathbb{R}^{d_x}$. Then, for the sequence $\{(x_k, y_k)\}$ derived from ZOB-GDA, we have for any $s \leq k$:*

$$\Delta_k \leq \alpha \Lambda_0^2 (2k - 2s + 1) + f(x_{k+1}, y_{k+1}) - f(x_k, y_k)$$
$$+ \frac{1}{2\beta} \left( \|y_k - y^*(x_s)\|^2 - \|y_{k+1} - y^*(x_s)\|^2 \right).$$

For the $j$th block, using the result in Lemma B.3 and letting $s = jB$, we have

$$\sum_{k=jB}^{(j+1)B-1} \Delta_k \leq \alpha \Lambda_0^2 (B^2 + B) + \frac{R_y^2}{2\beta} + \mathbb{E}\left[ f(x_{jB+B}, y_{jB+B}) - f(x_{jB}, y_{jB}) \right].$$

Substituting the above inequality with $j = 0, 1, \cdots, K/B - 1$ into (15), we have

$$\frac{1}{K} \sum_{k=0}^{K-1} \Delta_k \leq \alpha \Lambda_0^2 (B + 1) + \frac{R_y^2}{2\beta B} + \frac{1}{K} \mathbb{E}\left[ f(x_K, y_K) - f(x_0, y_0) \right].$$

We further have

$$\mathbb{E}\left[ f(x_K, y_K) - f(x_0, y_0) \right]$$
$$= \mathbb{E}\left[ f(x_K, y_K) - f(x_0, y_K) + f(x_0, y_K) - f(x_0, y_0) \right]$$
$$\leq \alpha \Lambda_0^2 K + \Delta_0.$$

Combining the above two inequalities leads to the result in Lemma B.2.

## B.3 Proof of Lemma B.3

Based on the definition of projection, we have for any $y \in \mathcal{Y}$

$$\langle y_{k+1} - y_k - \beta \nabla_y f(x_k, y_k), \ y - y_{k+1} \rangle \geq 0.$$

Rearranging this inequality, we can have

$$\frac{1}{2\beta} \left( \|y - y_k\|^2 - \|y - y_{k+1}\|^2 - \|y_{k+1} - y_k\|^2 \right)$$
$$\geq \langle y - y_{k+1}, \ \nabla_y f(x_k, y_k) \rangle \tag{16}$$
$$= \langle y - y_k, \ \nabla_y f(x_k, y_k) \rangle + \langle y_k - y_{k+1}, \ \nabla_y f(x_k, y_k) \rangle.$$

Using the concavity and smoothness of $f(x, y)$ in $y$, we have

$$\langle y - y_k, \ \nabla_y f(x_k, y_k) \rangle \geq f(x_k, y) - f(x_k, y_k),$$

$$f(x_k, y_{k+1}) - f(x_k, y_k) \geq \langle y_{k+1} - y_k, \ \nabla_y f(x_k, y_k) \rangle - \frac{L}{2} \|y_{k+1} - y_k\|^2.$$

Substituting the above two bounds into (16) and using the condition $\beta \leq \frac{1}{L}$, we have

$$f(x_k, y_{k+1}) - f(x_k, y) + \frac{1}{2\beta} \left( \|y - y_k\|^2 - \|y - y_{k+1}\|^2 \right) \geq 0. \tag{17}$$

Combining the definition of $\Delta_k$ with the above inequality with $y = y^*(x_s)$, we have

$$
\begin{aligned}
\Delta_k &= f(x_k, y^*(x_k)) - f(x_k, y_k) \\
&\leq f(x_k, y^*(x_k)) - f(x_k, y_k) + f(x_k, y_{k+1}) - f(x_k, y^*(x_s)) \\
&\quad + \frac{1}{2\beta} \left( \|y^*(x_s) - y_k\|^2 - \|y^*(x_s) - y_{k+1}\|^2 \right) \\
&= \underbrace{f(x_k, y^*(x_k)) - f(x_s, y^*(x_s))}_{E_1} + \underbrace{f(x_s, y^*(x_s)) - f(x_k, y^*(x_s))}_{E_2} \\
&\quad + \underbrace{f(x_k, y_{k+1}) - f(x_{k+1}, y_{k+1})}_{E_3} + \left( f(x_{k+1}, y_{k+1}) - f(x_k, y_k) \right) \\
&\quad + \frac{1}{2\beta} \left( \|y^*(x_s) - y_k\|^2 - \|y^*(x_s) - y_{k+1}\|^2 \right).
\end{aligned}
\tag{18}
$$

Due to that $f(x_s, y^*(x_k)) \leq f(x_s, y^*(x_s))$, we have

$$
\begin{aligned}
E_1 &\leq f(x_k, y^*(x_k)) - f(x_s, y^*(x_k)) \\
&\leq \Lambda \|x_k - x_s\| \\
&\leq \alpha \Lambda_0^2 (k - s).
\end{aligned}
$$

Similarly, we also have

$$
E_2 \leq \Lambda \|x_k - x_s\| \leq \alpha \Lambda_0^2 (k - s),
$$

and

$$
E_3 \leq \Lambda \|x_{k+1} - x_k\| \leq \alpha \Lambda_0^2.
$$

Substituting these bounds into (18) leads to the final result.

## C    PROOF OF THEOREM 4.1

We define some auxiliary notation:

$$
d(y, z) = \min_{x \in \mathbb{R}^{d_x}} K(x, y; z), \quad m(z) = \min_{x \in \mathbb{R}^{d_x}} \max_{y \in \mathcal{Y}} K(x, y; z),
$$

$$
h(x, z) = \max_{y \in \mathcal{Y}} K(x, y; z), \quad x(y, z) = \arg \min_{x \in \mathbb{R}^{d_x}} K(x, y; z),
$$

$$
x^*(z) = \arg \min_{x \in \mathbb{R}^{d_x}} h(x, z), \quad \mathcal{Y}(z) = \arg \max_{y \in \mathcal{Y}} d(y, z),
$$

$$
y_+(z_k) = \mathcal{P}_{\mathcal{Y}}[y_k + \beta \nabla_y K(x(y_k, z_k), y_k, z_k)].
$$

Note that $\mathcal{Y}(z)$ is a set, and we use $y(z)$ to denote an arbitrary element in $\mathcal{Y}(z)$. Recall that we assume $f(x, y)$ is $L$-smooth in $x$ and $y$. Then, if $p > L$, $K(x, y; z)$ is $(p - L)$-strongly convex in $x$ and smooth in $x$ with a constant $(L + p)$. We define the potential function:

$$
\phi(x, y, z) = K(x, y; z) - 2d(y, z) + 2m(z).
$$

For simplicity, we denote $\phi_k = \phi(x_k, y_k; z_k)$.

Before providing our formal proof, we present some supporting lemmas.

### C.1    SUPPORTING LEMMAS FOR THEOREM 4.1

**Lemma C.1.** *For any $x, z \in \mathbb{R}^{d_x}$ and $y \in \mathcal{Y}$, $\phi(x, y; z)$ is lower bounded by $\underline{f}$.*

*Proof of Lemma C.1.* We have

$$
\phi(x, y; z) = m(z) + (K(x, y; z) - d(y, z)) + (m(z) - d(y, z)) \geq m(z) \geq \underline{f},
$$

where the first inequality follows from the definition of $d(y, z)$ and $m(z)$. The second one holds because $\Phi(x) = \max_{y \in \mathcal{Y}} f(x, y)$ is lower bounded by $\underline{f}$.

$\square$

**Lemma C.2.** *There exists some constants $\sigma_1, \sigma_2$ satisfying*

$$\|x(y, z) - x(y, z')\| \leq \sigma_1 \|z - z'\|,$$
$$\|x^*(z) - x^*(z')\| \leq \sigma_1 \|z - z'\|,$$
$$\|x(y, z) - x(y', z)\| \leq \sigma_2 \|y - y'\|,$$

*for any $y, y' \in \mathcal{Y}$ and $z, z' \in \mathbb{R}^{d_x}$, where $\sigma_1 = \frac{p}{p-L}, \sigma_2 = \frac{2(p+L)}{p-L}$.*

Lemma C.2 follows from the results in (Zhang et al., 2020, Lemma B.2). Therefore, we omit its proof here.

**Lemma C.3.** *The dual function $d(y, z)$ is differentiable in $y$ and $L_d$-smooth in $y$, i.e., $\|\nabla_y d(y, z) - \nabla_y d(y', z)\| \leq L_d \|y - y'\|, \forall y, y' \in \mathcal{Y}$, where $L_d = L + L\sigma_2$.*

*Proof of Lemma C.3.* Based on Danskin's Theorem, we have $\nabla_y d(y, z) = \nabla_y K(x(y, z), y; z) = \nabla_y f(x(y, z), y)$. Then, we have for any $y, y' \in \mathcal{Y}$

$$\|\nabla_y d(y, z) - \nabla_y d(y', z)\|$$
$$= \|\nabla_y K(x(y, z), y; z) - \nabla_y K(x(y', z), y'; z)\|$$
$$\leq \|\nabla_y K(x(y, z), y; z) - \nabla_y K(x(y, z), y'; z)\|$$
$$\quad + \|\nabla_y K(x(y, z), y'; z) - \nabla_y K(x(y', z), y'; z)\|$$
$$\leq L\|y - y'\| + L\|x(y, z) - x(y', z)\|$$
$$\leq (L + \sigma_2 L)\|y - y'\|,$$

where the third step follows from the $L$-smoothness of $K(x, y; z)$ in $y$, and the last step follows from Lemma C.2. $\qquad\square$

**Lemma C.4.** *For the sequence $\{(x_k, y_k, z_k)\}$ derived from ZOB-SGDA, we have*

$$\mathbb{E}\left[\|x_{k+1} - x(y_k, z_k)\|^2\right] \leq 4\sigma_3^2 \mathbb{E}\left[\|x_{k+1} - x_k\|^2\right] + \frac{L^2 r_k^2 d_x}{(p-L)^2}, \tag{19}$$

*where $\sigma_3 = \frac{\sqrt{N} + \alpha(p-L)}{\alpha(p-L)}$.*

*Proof of Lemma C.4.* Denote $\mathcal{F}_k = \sigma(x_0, y_0, \mathcal{I}_0, \cdots, i_{k-1}, x_k, y_k)$ as a filtration. By Lemma 3.10 in Zhang & Luo (2020), we have

$$\|x_k - x(y_k, z_k)\| \leq \frac{1}{\alpha(p-L)} \|x_k - \mathcal{P}_{\mathcal{X}}[x_k - \alpha \nabla_x K(x_k, y_k; z_k)]\|$$
$$= \frac{1}{\alpha(p-L)} \|\alpha \nabla_x K(x_k, y_k; z_k)\|, \tag{20}$$

where $\mathcal{X} = \mathbb{R}^{d_x}$ in our algorithm. Then, we can get

$$\|x_{k+1} - x(y_k, z_k)\|^2$$
$$\leq 2\|x_{k+1} - x_k\|^2 + 2\|x_k - x(y_k, z_k)\|^2$$
$$\leq 2\|x_{k+1} - x_k\|^2 + \frac{2}{\alpha^2 (p-L)^2} \|\alpha \nabla_x K(x_k, y_k; z_k)\|^2$$
$$\leq 2\|x_{k+1} - x_k\|^2 + \frac{4}{\alpha^2 (p-L)^2} \|\alpha G_x(x_k, y_k; z_k)\|^2$$
$$\quad + \frac{4}{(p-L)^2} \|\nabla_x K(x_k, y_k, z_k) - G_x(x_k, y_k; z_k)\|^2$$
$$\leq 2\|x_{k+1} - x_k\|^2 + \frac{4N}{\alpha^2 (p-L)^2} \mathbb{E}\left[\|x_k - x_{k+1}\|^2 \,\Big|\, \mathcal{F}_k\right] + \frac{L^2 r_k^2 d_x}{(p-L)^2},$$

where the first and third steps follow from the Cauchy-Schwarz inequality. The second step follows from Eq.(20). In the last step, we applied Lemma 3.1 and $\mathbb{E}\left[\|\alpha G_x(x_k, y_k; z_k)\|^2\right] = N\mathbb{E}\left[\|x_{k+1} - x_k\|^2\right]$. Taking the expectation of both sides of the above inequality leads to

$$\mathbb{E}\left[\|x_{k+1} - x(y_k, z_k)\|^2\right] \leq \left(2 + \frac{4N}{\alpha^2(p-L)^2}\right)\mathbb{E}\left[\|x_{k+1} - x_k\|^2\right] + \frac{L^2 r_k^2 d_x}{(p-L)^2}$$

$$\leq 4\sigma_3^2 \mathbb{E}\left[\|x_{k+1} - x_k\|^2\right] + \frac{L^2 r_k^2 d_x}{(p-L)^2}.$$

$\square$

**Lemma C.5.** *For any $k \geq 0$, we have*

$$\mathbb{E}\left[\|y_{k+1} - y_+(z_k)\|^2\right] \leq \kappa \mathbb{E}\left[\|x_{k+1} - x_k\|^2\right] + \frac{\beta^2 L^2 r_k^2 d_x}{2},$$

*where $\kappa = (8\sigma_3^2 + 2)\beta^2 L^2$.*

*Proof of Lemma C.5.* By the non-expansiveness of the projection operator, we have

$$\|y_{k+1} - y_+(z_k)\|^2$$
$$= \|\mathcal{P}_{\mathcal{Y}}[y_k - \beta \cdot \nabla_y K(x_k, y_k; z_k)] - \mathcal{P}_{\mathcal{Y}}[y_k - \beta \cdot \nabla_y K(x(y_k, z_k), y_k; z_k)]\|^2$$
$$\leq \beta^2 \|\nabla_y K(x(y_k, z_k), y_k; z_k) - \nabla_y K(x_k, y_k; z_k)\|^2$$
$$\leq \beta^2 L^2 \|x_k - x(y_k, z_k)\|^2.$$

where in the first step we used the non-expansiveness of projection operations and in the third step we used the $L$-smoothness of $K(x, y; z)$ in $x$. Then, taking the expectation of the above inequality leads to

$$\mathbb{E}\left[\|y_{k+1} - y_+(z_k)\|^2\right]$$
$$\leq \beta^2 L^2 \mathbb{E}\left[\|x_k - x(y_k, z_k)\|^2\right]$$
$$\leq 2\beta^2 L^2 \mathbb{E}\left[\|x_{k+1} - x_k\|^2\right] + 2\beta^2 L^2 \mathbb{E}\left[\|x_{k+1} - x(y_k, z_k)\|^2\right]$$
$$\leq \left(8\beta^2 L^2 \sigma_3^2 + 2\beta^2 L^2\right)\mathbb{E}\left[\|x_{k+1} - x_k\|^2\right] + \frac{2\beta^2 L^4 r_k^2 d_x}{(p-L)^2}$$
$$\leq \left(8\beta^2 L^2 \sigma_3^2 + 2\beta^2 L^2\right)\mathbb{E}\left[\|x_{k+1} - x_k\|^2\right] + \frac{\beta^2 L^2 r_k^2 d_x}{2},$$

where in the second step we applied Lemma C.4 and in the last step we applied the inequality (19) and the condition $p \geq 3L$. $\square$

**Lemma C.6.** *For any $k \geq 0$, we have*

$$\beta(p-L)\|x^*(z_k) - x(y_+(z_k), z_k)\|^2 \leq (1 + \beta L + \beta L\sigma_2)D_y\|y_k - y_+(z_k)\|.$$

Lemma C.6 comes from (Zhang et al., 2020, Lemma B.10).

## C.2 FORMAL PROOF OF THEOREM 4.1

In this subsection, we provide the formal proof of Theorem 4.1. The proof mainly contains three steps as follows.

**Step 1: Derive a Bound for the Stationarity Measure.** Recall that the stationarity measure is defined as $M(x, y) = \min\{\|\Phi(x)\|, \|\mathfrak{g}(x, y)\|\}$. Here we analyze the convergence of $\|\mathfrak{g}(x, y)\|$. Also, recall the definition of $\mathfrak{g}(x, y)$:

$$\mathfrak{g}(x, y) = \begin{pmatrix} \mathfrak{g}_x(x, y) \\ \mathfrak{g}_y(x, y) \end{pmatrix} = \begin{pmatrix} \nabla_x f(x, y) \\ \frac{1}{\beta}\left(y - \mathcal{P}_{\mathcal{Y}}\left[y + \beta \nabla_y f(x, y)\right]\right) \end{pmatrix},$$

for any $(x, y) \in \mathbb{R}^{d_x} \times \mathcal{Y}$. Then, we provide a bound on the stationarity measure in the following lemma.

**Lemma C.7.** *For any $\{x_k, y_k, z_k\}_{k \geq 0}$ derived from ZOB-SGDA, we have*

$$
\mathbb{E}\left[\|\mathfrak{g}(x_k, y_k)\|^2\right] \leq \mathbb{E}\left[\left(\frac{3N}{\alpha^2} + 8L^2\sigma_3^2 + 6p^2\right)\|x_{k+1} - x_k\|^2\right] + \frac{5L^2 r_k^2 d_x}{4}
$$
$$
+ \mathbb{E}\left[\frac{2}{\beta^2}\|y_k - y_+(z_k)\|^2 + 6p^2\|x_{k+1} - z_k\|^2\right].
$$

*Proof of Lemma C.7.* Based on the update of $x$ : $x_{k+1} = x_k - \alpha \cdot G_x^{\mathcal{I}_k}(x_k, y_k; z_k)$, we have

$$
\mathbb{E}\left[\|\mathfrak{g}_x(x_k, y_k)\|^2\right]
$$
$$
= \mathbb{E}\left[\|G_x(x_k, y_k; z_k) + \nabla_x K(x_k, y_k z_k) - G_x(x_k, y_k; z_k) - p(x_k - z_k)\|^2\right]
$$
$$
\leq \mathbb{E}\left[3\|G_x(x_k, y_k; z_k)\|^2 + 3\|G_x(x_k, y_k; z_k) - \nabla_x K(x_k, y_k; z_k)\|^2 + 3p^2\|x_k - z_k\|^2\right]
$$
$$
\leq \mathbb{E}\left[\frac{3N}{\alpha^2}\|x_{k+1} - x_k\|^2 + 3p^2\|x_k - z_k\|^2\right] + \frac{3L^2 r_k^2 d_x}{4}
$$
$$
\leq \mathbb{E}\left[\left(\frac{3N}{\alpha^2} + 6p^2\right)\|x_{k+1} - x_k\|^2 + 6p^2\|x_{k+1} - z_k\|^2\right] + \frac{3L^2 r_k^2 d_x}{4},
$$

where in the third step we applied the relation

$$
\mathbb{E}\left[\frac{3N}{\alpha^2}\|x_{k+1} - x_k\|^2\right] = \mathbb{E}\left[3N\|G_x^{\mathcal{I}_k}(x_k, y_k)\|^2\right] = \mathbb{E}\left[3\|G_x(x_k, y_k)\|^2\right].
$$

For the term $\|\mathfrak{g}_y(x_k, y_k)\|^2$, we can get

$$
\mathbb{E}\left[\|\mathfrak{g}_y(x_k, y_k)\|^2\right]
$$
$$
= \frac{1}{\beta^2}\mathbb{E}\left[\|y_{k+1} - y_k\|^2\right]
$$
$$
\leq \frac{2}{\beta^2}\mathbb{E}\left[\|y_{k+1} - y_+(z_k)\|^2\right] + \frac{2}{\beta^2}\mathbb{E}\left[\|y_+(z_k) - y_k\|^2\right]
$$
$$
\leq \frac{8\kappa^2}{\beta^2}\mathbb{E}\left[\|x_{k+1} - x_k\|^2\right] + \frac{2}{\beta^2}\mathbb{E}\left[\|y_+(z_k) - y_k\|^2\right] + \frac{L^2 r_k^2 d_x}{2}
$$
$$
= 8L^2\sigma_3^2\mathbb{E}\left[\|x_{k+1} - x_k\|^2\right] + \frac{2}{\beta^2}\mathbb{E}\left[\|y_+(z_k) - y_k\|^2\right] + \frac{L^2 r_k^2 d_x}{2},
$$

where we applied Lemma C.5 in the third step. Finally, combining it with the bound on $\mathbb{E}[\|\mathfrak{g}_x(x_k, y_k)\|^2]$ leads to the desired result. $\qquad\square$

**Step 2: Derive a bound on the one-step drift of potential function.** We provide a bound on the one-step drift in the following lemma.

**Lemma C.8.** *Suppose the assumptions and conditions in Theorem 4.1 hold. For any $\{x_k, y_k, z_k\}$ derived from ZOB-SGDA, we have*

$$
\mathbb{E}\left[\phi_k - \phi_{k+1}\right]
$$
$$
\geq \mathbb{E}\left[\underbrace{\frac{1}{8\alpha}\|x_{k+1} - x_k\|^2 + \frac{1}{8\beta}\|y_k - y_+(z_k)\|^2 + \frac{p}{8\gamma}\|z_{k+1} - z_k\|^2}_{T_1}\right]
$$
$$
\tag{21}
$$
$$
- \mathbb{E}\left[\underbrace{24p\gamma\|x^*(z_k) - x(y_+(z_k), z_k)\|^2}_{T_2}\right] - \frac{L^3 r_k^2 \alpha^2 d_x}{N}
$$
$$
- 12\beta^2 L^2 r_k^2 \sigma_2^2 p\gamma d_x - \frac{\beta L^2 r_k^2 d_x}{8} - \frac{L^2 r_k^2 d_x}{8N}.
$$

*Proof of Lemma C.8.* First, we provide the following lemma to characterize the descent in primal steps.

**Lemma C.9.** *For any $k \geq 0$, we have*

$$K(x_k, y_k; z_k) - K(x_{k+1}, y_{k+1}; z_{k+1})$$

$$\geq \left(\frac{1}{\alpha} - \frac{p+L+1}{2}\right) \|x_{k+1} - x_k\|^2 - \frac{L}{2}\|y_k - y_{k+1}\|^2 + \frac{p}{2\gamma}\|z_{k+1} - z_k\|^2$$

$$+ \langle \nabla_y K(x_{k+1}, y_k; z_k), \ y_k - y_{k+1} \rangle - \frac{L^2 r_k^2 d_x}{8N}.$$

*Proof of Lemma C.9.* By the update of $x$, we use the smoothness of $K$ to get

$$K(x_{k+1}, y_k; z_k) - K(x_k, y_k; z_k)$$

$$\leq \langle \nabla_x K(x_k, y_k; z_k), \ x_{k+1} - x_k \rangle + \frac{p+L}{2}\|x_{k+1} - x_k\|^2$$

$$= \langle G_x(x_k, y_k; z_k), \ x_{k+1} - x_k \rangle + \frac{p+L}{2}\|x_{k+1} - x_k\|^2$$

$$+ \langle \nabla_x K(x_k, y_k; z_k) - G_x(x_k, y_k; z_k), \ x_{k+1} - x_k \rangle \qquad (22)$$

$$\leq \langle G_x(x_k, y_k; z_k), \ x_{k+1} - x_k \rangle + \frac{p+L+1}{2}\|x_{k+1} - x_k\|^2 + \frac{L^2 r_k^2 d_x}{8N}$$

$$\leq \left(-\frac{1}{\alpha} + \frac{p+L+1}{2}\right) \|x_{k+1} - x_k\|^2 + \frac{L^2 r_k^2 d_x}{8N},$$

where in the second inequality we applied AM-GM inequality and the fact that only the entries of $\mathcal{I}_k$ in $x_{k+1} - x_k$ are nonzero. Similarly, we can use the $L$-smoothness of $K(x, y; z)$ to get

$$K(x_{k+1}, y_k; z_k) - K(x_{k+1}, y_{k+1}; z_k)$$

$$\geq \langle \nabla_y K(x_{k+1}, y_k; z_k), \ y_k - y_{k+1} \rangle - \frac{L}{2}\|y_{k+1} - y_k\|^2. \qquad (23)$$

Based on the update of $z$, we have

$$K(x_{k+1}, y_{k+1}; z_k) - K(x_{k+1}, y_{k+1}; z_{k+1}) \geq \frac{p}{2\gamma}\|z_{k+1} - z_k\|^2. \qquad (24)$$

Combining the results in (22), (23), and (24) leads to the final result. $\qquad \square$

We further provide the following two lemmas to characterize the one-step drift of $d(y, z)$ and $m(z)$ in ZOB-SGDA. Their proofs follow from (Zhang et al., 2020, Lemma B.6 and Lemma B.7).

**Lemma C.10.** *For any $k$, we have*

$$d(y_{k+1}, z_{k+1}) - d(y_k, z_k)$$

$$\geq \langle \nabla_y K(x(y_k, z_k), y_k; z_k), \ y_{k+1} - y_k \rangle - \frac{L_d}{2}\|y_{k+1} - y_k\|^2$$

$$+ \frac{p}{2}\langle z_{k+1} - z_k, \ z_{k+1} + z_k - 2x(y_{k+1}, z_{k+1})\rangle.$$

*Proof of Lemma C.10.* Using the smoothness of $d(y, z)$ in $y$ provided in Lemma C.3, we have

$$d(y_{k+1}, z_k) - d(y_k, z_k)$$

$$\geq \langle \nabla_y d(y_k, z_k), y_{k+1} - y_k \rangle - \frac{L_d}{2}\|y_{k+1} - y_k\|^2$$

$$= \langle \nabla_y K(x(y_k, z_k), y_k; z_k), y_{k+1} - y_k \rangle - \frac{L_d}{2}\|y_{k+1} - y_k\|^2,$$

where in the second step we used $\nabla_y d(y_k, z_k) = \nabla_y K(x(y_k, z_k), y_k; z_k)$. Also, we have

$$
\begin{aligned}
&d(y_{k+1}, z_{k+1}) - d(y_{k+1}, z_k) \\
&= K(x(y_{k+1}, z_{k+1}), y_{k+1}; z_{k+1}) - K(x(y_{k+1}, z_k), y_{k+1}; z_k) \\
&\geq K(x(y_{k+1}, z_{k+1}), y_{k+1}; z_{k+1}) - K(x(y_{k+1}, z_{k+1}), y_{k+1}; z_k) \\
&= \frac{p}{2}\|x_{k+1} - z_{k+1}\|^2 - \frac{p}{2}\|x_{k+1} - z_k\|^2 \\
&= \frac{p}{2}\langle z_{k+1} - z_k, \ z_{k+1} + z_k - 2x(y_{k+1}, z_{k+1})\rangle.
\end{aligned}
$$

Finally, combining the above two inequalities leads to the desired result.

$\square$

**Lemma C.11.** *For any k, we have*

$$
m(z_{k+1}) - m(z_k) \leq \frac{p}{2}\langle z_{k+1} - z_k, \ z_{k+1} + z_k - 2x(y(z_{k+1}), z_k)\rangle,
$$

*where $y(z_{k+1})$ is an arbitrary element in $\mathcal{Y}(z_{k+1})$.*

*Proof.* Using Kakutoni's Theorem, we have

$$
m(z) = \max_{y \in \mathcal{Y}} d(y, z) = d(y(z), z).
$$

Therefore, we have

$$
\begin{aligned}
&m(z_{k+1}) - m(z_k) \\
&\leq d(y(z_{k+1}), z_{k+1}) - d(y(z_{k+1}), z_k) \\
&= K(x(y(z_{k+1}), z_{k+1}), y(z_{k+1}); z_{k+1}) - K(x(y(z_{k+1}), z_k), y(z_{k+1}); z_k) \\
&\leq K(x(y(z_{k+1}), z_k), y(z_{k+1}); z_{k+1}) - K(x(y(z_{k+1}), z_k), y(z_{k+1}); z_k) \\
&= \frac{p}{2}\langle z_{k+1} - z_k, \ z_{k+1} + z_k - 2x(y(z_{k+1}), z_k)\rangle,
\end{aligned}
$$

where in the first step and third step we used the definitions of $y(z)$ and $x(y, z)$, respectively. $\square$

Now we can bound the one-step drift of the potential function. Using the results in Lemmas C.9, C.10, and C.11, we have

$$
\begin{aligned}
&\phi_k - \phi_{k+1} \\
&\geq \left(\frac{1}{\alpha} - \frac{p + L + 1}{2}\right)\|x_{k+1} - x_k\|^2 - \frac{L + 2L_d}{2}\|y_{k+1} - y_k\|^2 \\
&\quad + \frac{p}{2\gamma}\|z_{k+1} - z_k\|^2 + \langle \nabla_y K(x_{k+1}, y_k; z_k), y_{k+1} - y_k\rangle \\
&\quad + 2\langle \nabla_y K(x(y_k, z_k), y_k; z_k) - \nabla_y K(x_{k+1}, y_k; z_k), \ y_{k+1} - y_k\rangle \\
&\quad + 2p(z_{k+1} - z_k)^T (x(y(z_{k+1}), z_k) - x(y_{k+1}, z_{k+1})) - \frac{L^2 r_k^2 d_x}{8N}.
\end{aligned}
$$

Using the property of the projection operator:

$$
\langle y_{k+1} - (y_k + \beta \nabla_y K(x_k, y_k; z_k)), \ y_{k+1} - y\rangle \leq 0,
$$

for any $y \in \mathcal{Y}$, and setting $y = y_k$, we have

$$
\langle \nabla_y K(x_k, y_k; z_k), y_{k+1} - y_k\rangle \geq \frac{1}{\beta}\|y_{k+1} - y_k\|^2.
$$

Therefore,

$$
\begin{aligned}
&\langle \nabla_y K(x_{k+1}, y_k; z_k), y_{k+1} - y_k\rangle \\
&= \langle \nabla_y K(x_k, y_k; z_k), y_{k+1} - y_k\rangle + \langle \nabla_y K(x_{k+1}, y_k; z_k) - \nabla_y K(x_k, y_k; z_k), y_{k+1} - y_k\rangle \\
&\geq \frac{1}{\beta}\|y_{k+1} - y_k\|^2 - \frac{L}{2}\|y_{k+1} - y_k\|^2 - \frac{L}{2}\|x_{k+1} - x_k\|^2 \\
&= \left(\frac{1}{\beta} - \frac{L}{2}\right)\|y_{k+1} - y_k\|^2 - \frac{L}{2}\|x_{k+1} - x_k\|^2,
\end{aligned}
$$

where the first inequality follows from AM-GM inequality and the smoothness of $K$. Then, we can further get

$$
\begin{aligned}
& \phi_k - \phi_{k+1} \\
& \geq \left( \frac{1}{\alpha} - \frac{p+2L+1}{2} \right) \|x_{k+1} - x_k\|^2 + \left( \frac{1}{\beta} - (L + L_d) \right) \|y_{k+1} - y_k\|^2 + \frac{p}{2\gamma} \|z_{k+1} - z_k\|^2 \\
& \quad + 2\langle \nabla_y K(x(y_k, z_k), y_k; z_k) - \nabla_y K(x_{k+1}, y_k; z_k),\ y_{k+1} - y_k \rangle \\
& \quad + 2p(z_{k+1} - z_k)^T \left( x(y(z_{k+1}), z_k) - x(y_{k+1}, z_{k+1}) \right) - \frac{L^2 r_k^2 d_x}{8N}.
\end{aligned}
\tag{25}
$$

Besides, we have

$$
\begin{aligned}
& 2p(z_{k+1} - z_k)^T \left( x(y(z_{k+1}), z_k) - x(y_{k+1}, z_{k+1}) \right) \\
& = 2p(z_{k+1} - z_k)^T \left( x(y(z_{k+1}), z_k) - x(y(z_{k+1}), z_{k+1}) \right) \\
& \quad + 2p(z_{k+1} - z_k)^T \left( x(y(z_{k+1}), z_{k+1}) - x(y_{k+1}, z_{k+1}) \right) \\
& \geq -2p\sigma_1 \|z_{k+1} - z_k\|^2 + 2p(z_{k+1} - z_k)^T \left( x(y(z_{k+1}), z_{k+1}) - x(y_{k+1}, z_{k+1}) \right) \\
& \geq -2p\sigma_1 \|z_{k+1} - z_k\|^2 - \frac{p}{6\gamma} \|z_{k+1} - z_k\|^2 - 6p\gamma \|x(y(z_{k+1}), z_{k+1}) - x(y_{k+1}, z_{k+1})\|^2,
\end{aligned}
\tag{26}
$$

where the second step follows from Lemma C.2 and the third step follows from the AM-GM inequality. We also have the bound:

$$
\begin{aligned}
& \mathbb{E} \left[ 2\langle \nabla_y K(x(y_k, z_k), y_k; z_k) - \nabla_y K(x_{k+1}, y_k; z_k),\ y_{k+1} - y_k \rangle \right] \\
& \geq \mathbb{E} \left[ -2L \|x_{k+1} - x(y_k, z_k)\| \cdot \|y_{k+1} - y_k\| \right] \\
& \geq \mathbb{E} \left[ -L\sigma_3^2 \|y_{k+1} - y_k\|^2 - L\sigma_3^{-2} \|x_{k+1} - x(y_k, z_k)\|^2 \right] \\
& \geq \mathbb{E} \left[ -L\sigma_3^2 \|y_{k+1} - y_k\|^2 - 4L \|x_{k+1} - x_k\|^2 \right] - \frac{L^3 r_k^2 \alpha^2 d_x}{\left( \sqrt{N} + \alpha(p-L) \right)^2} \\
& \geq \mathbb{E} \left[ -L\sigma_3^2 \|y_{k+1} - y_k\|^2 - 4L \|x_{k+1} - x_k\|^2 \right] - \frac{L^3 r_k^2 \alpha^2 d_x}{N},
\end{aligned}
\tag{27}
$$

where the first step follows from the smoothness of $K(x, y; z)$ and the third step follows from Lemma C.4. In the last step, we used the fact that $p > L$. Taking the expectation on both sides of (25) and combining it with the bounds in (26) and (27), we can get

$$
\begin{aligned}
& \mathbb{E} \left[ \phi_k - \phi_{k+1} \right] \\
& \geq \mathbb{E} \left[ \left( \frac{1}{\alpha} - \frac{p+2L+1}{2} - 4L \right) \|x_{k+1} - x_k\|^2 + \left( \frac{1}{\beta} - L - L_d - L\sigma_3^2 \right) \|y_{k+1} - y_k\|^2 \right] \\
& \quad + \mathbb{E} \left[ \left( \frac{p}{2\gamma} - 2p\sigma_1 - \frac{p}{6\gamma} \right) \|z_{k+1} - z_k\|^2 \right] \\
& \quad - \mathbb{E} \left[ 6p\gamma \|x(y(z_{k+1}), z_{k+1}) - x(y_{k+1}, z_{k+1})\|^2 \right] - \frac{L^3 r_k^2 \alpha^2 d_x}{N} - \frac{L^2 r_k^2 d_x}{8N}.
\end{aligned}
\tag{28}
$$

Using the conditions $\beta \leq \min \left\{ \frac{1}{12L}, \frac{\alpha^2 (p-L)^2}{4L(\sqrt{N} + \alpha(p-L))^2} \right\}$, we have

$$
L + L_d \leq 6L \leq \frac{1}{2\beta},
$$

and

$$
L\sigma_3^2 = \frac{L(\sqrt{N} + \alpha(p-L))^2}{\alpha^2 (p-L)^2} \leq \frac{1}{4\beta}.
$$

Therefore, we have

$$
\frac{1}{\beta} - L - L_d - L\sigma_3^2 \geq \frac{1}{4\beta}.
\tag{29}
$$

Using Lemma C.5, we have the bound on $\mathbb{E}[\|y_{k+1} - y_k\|^2]$:

$$\mathbb{E}\left[\|y_{k+1} - y_k\|^2\right]$$
$$\geq \mathbb{E}\left[\frac{1}{2}\|y_k - y_+(z_k)\|^2 - \|y_{k+1} - y_+(z_k)\|^2\right]$$
$$\geq \mathbb{E}\left[\frac{1}{2}\|y_k - y_+(z_k)\|^2 - \kappa\|x_{k+1} - x_k\|^2\right] - \frac{\beta^2 L^2 r_k^2 d_x}{2}.$$

As for the bound on $\mathbb{E}[\|x^*(z_{k+1}) - x(y_{k+1}, z_{k+1})\|^2]$, we have

$$\mathbb{E}\left[\|x^*(z_{k+1}) - x(y_{k+1}, z_{k+1})\|^2\right]$$
$$\leq \mathbb{E}\left[2\|x^*(z_{k+1}) - x^*(z_k) + x^*(z_k) - x(y_+(z_k), z_k)\|^2\right.$$
$$\left. +2\|x(y_+(z_k), z_k) - x(y_{k+1}, z_k) + x(y_{k+1}, z_k) - x(y_{k+1}, z_{k+1})\|^2\right]$$
$$\leq \mathbb{E}\left[4\|x^*(z_{k+1}) - x^*(z_k)\|^2 + 4\|x^*(z_k) - x(y_+(z_k), z_k)\|^2\right]$$
$$+ \mathbb{E}\left[4\|x(y_+(z_k), z_k) - x(y_{k+1}, z_k)\|^2 + 4\|x(y_{k+1}, z_k) - x(y_{k+1}, z_{k+1})\|^2\right]$$
$$\leq \mathbb{E}\left[4\sigma_1^2\|z_{k+1} - z_k\|^2 + 4\|x^*(z_k) - x(y_+(z_k), z_k)\|^2\right]$$
$$+ \mathbb{E}\left[4\sigma_2^2\|y_+(z_k) - y_{k+1}\|^2 + 4\sigma_1^2\|z_k - z_{k+1}\|^2\right]$$
$$\leq \mathbb{E}\left[8\sigma_1^2\|z_{k+1} - z_k\|^2 + 4\|x^*(z_k) - x(y_+(z_k), z_k)\|^2\right.$$
$$\left. +4\sigma_2^2\kappa\|x_{k+1} - x_k\|^2\right] + 2\beta^2 L^2 r_k^2 \sigma_2^2 d_x,$$

where the first two steps follow from the Cauchy-Schwarz inequality and the last two steps follow from Lemma C.2 and C.5. Combining the above two bounds with (29) and (28) leads to

$$\mathbb{E}\left[\phi_k - \phi_{k+1}\right]$$
$$\geq \mathbb{E}\left[\left(\frac{1}{\alpha} - \frac{p + 2L + 1}{2} - 4L - \frac{\kappa}{4\beta} - 24p\gamma\sigma_2^2\kappa\right)\|x_{k+1} - x_k\|^2\right]$$
$$+ \mathbb{E}\left[\frac{1}{8\beta}\|y_k - y_+(z_k)\|^2 + \left(\frac{p}{2\gamma} - 2p\sigma_1 - \frac{p}{6\gamma} - 48p\gamma\sigma_1^2\right)\|z_{k+1} - z_k\|^2\right]$$
$$- \mathbb{E}\left[24p\gamma\|x^*(z_k) - x(y_+(z_k), z_k)\|^2\right]$$
$$- \frac{L^3 r_k^2 \alpha^2 d_x}{N} - 12p\gamma d_x \sigma_2^2 \beta^2 L^2 r_k^2 - \frac{\beta L^2 r_k^2 d_x}{8} - \frac{L^2 r_k^2 d_x}{8N}.$$

Based on the condition $\alpha \leq \frac{1}{p+10L+1}$, we have $\alpha < \frac{1}{p+10L} \leq \frac{1}{13L}$ and $\frac{p+10L+1}{2} \leq \frac{1}{2\alpha}$. Using the condition $\beta \leq \min\left\{\frac{1}{12L}, \frac{\alpha^2(p-L)^2}{4L(\sqrt{N}+\alpha(p-L))^2}\right\}$, we have

$$\frac{\kappa}{4\beta} = 2\beta L^2 \sigma_3^2 + \frac{1}{2}\beta L^2 < L < \frac{1}{8\alpha}.$$

Furthermore, using the condition $\gamma \leq \frac{1}{768p\beta}$, we can get

$$24p\gamma\sigma_2^2\kappa \leq \frac{\sigma_2^2\kappa}{32\beta} \leq \frac{\kappa}{2\beta} \leq \frac{1}{4\alpha}.$$

Therefore, we have

$$\frac{1}{\alpha} - \frac{p + 2L + 1}{2} - 4L - \frac{\kappa}{4\beta} - 24p\gamma\sigma_2^2\kappa > \frac{1}{8\alpha}.$$

Using the condition $\gamma \leq \frac{1}{36}$, we have $2p\sigma_1 \leq 3p \leq \frac{p}{12\gamma}$ and $48p\gamma\sigma_1^2 \leq 3p \leq \frac{p}{12\gamma}$. Thus, we have

$$\frac{p}{2\gamma} - 2p\sigma_1 - \frac{p}{6\gamma} - 48p\gamma\sigma_1^2 > \frac{p}{8\gamma}.$$

Then, we can combine the above results to get

$$\mathbb{E}\left[\phi_k - \phi_{k+1}\right]$$

$$\geq \mathbb{E}\left[\frac{1}{8\alpha}\|x_{k+1} - x_k\|^2 + \frac{1}{8\beta}\|y_k - y_+(z_k)\|^2 + \frac{p}{8\gamma}\|z_{k+1} - z_k\|^2\right]$$

$$- \mathbb{E}\left[24p\gamma\|x^*(z_k) - x(y_+(z_k), z_k)\|^2\right] - \frac{L^3 r_k^2 \alpha^2 d_x}{N}$$

$$- 12p\gamma\beta^2 L^2 r_k^2 \sigma_2^2 d_x - \frac{\beta L^2 r_k^2 d_x}{8} - \frac{L^2 r_k^2 d_x}{8N}.$$

$$\square$$

**Step 3: Combine the above two bounds to get the convergence rate.** Then, we are ready to prove the final result. We consider two situations in (21): (1) $\frac{1}{2}T_1 \leq T_2$, (2) $\frac{1}{2}T_1 > T_2$. In the first case, we have

$$\frac{1}{16\alpha}\|x_{k+1} - x_k\|^2 + \frac{1}{16\beta}\|y_k - y_+(z_k)\|^2 + \frac{p}{16\gamma}\|z_{k+1} - z_k\|^2$$

$$\leq 24p\gamma\|x^*(z_k) - x(y_+(z_k), z_k)\|^2.$$

Let $t_1 = 384pD_y\frac{1+\beta L + \beta L\sigma_2}{p - L}$. Then, using Lemma C.6, we have

$$\|y_k - y_+(z_k)\|^2$$

$$\leq 384p\gamma\beta\|x^*(z_k) - x(y_+(z_k), z_k)\|^2$$

$$\leq 384p\gamma\beta D_y\frac{1 + \beta L + \beta L\sigma_2}{\beta(p - L)}\|y_k - y_+(z_k)\|,$$

which leads to $\|y_k - y_+(z_k)\| \leq t_1\gamma$. Therefore, we can get

$$\|z_{k+1} - z_k\|^2$$

$$\leq 384\gamma^2\|x^*(z_k) - x(y_+(z_k), z_k)\|^2$$

$$\leq 384\gamma^2 D_y\frac{1 + \beta L + \beta L\sigma_2}{\beta(p - L)}\|y_k - y_+(z_k)\|$$

$$= \frac{t_1^2\gamma^3}{p\beta},$$

and

$$\|x_{k+1} - x_k\|^2$$

$$\leq 384p\alpha\gamma\|x^*(z_k) - x(y_+(z_k), z_k)\|^2$$

$$\leq 384p\alpha\gamma D_y\frac{1 + \beta L + \beta L\sigma_2}{\beta(p - L)}\|y_k - y_+(z_k)\|$$

$$\leq \frac{\alpha t_1^2\gamma^2}{\beta}.$$

Combining the above three inequalities with Lemma C.7, we can get

$$\frac{1}{K}\sum_{k=0}^{K-1}\mathbb{E}\left[\|\mathfrak{g}(x_k, y_k)\|^2\right]$$

$$\leq \left(\frac{3N}{\alpha^2} + 8L^2\sigma_3^2 + 6p^2\right)\frac{\alpha t_1^2\gamma^2}{\beta} + \frac{2t_1\gamma^2}{\beta^2} + \frac{6p^2 t_1^2\gamma}{p\beta} + \frac{5L^2 d_x\sum_{k=0}^{K-1}r_k^2}{4K}.$$

By $\gamma \leq \frac{1}{\sqrt{KN}}$ and $\sum_{k=0}^{K-1}r_k^2 = \frac{1}{b}$, we can get $\frac{1}{K}\sum_{k=0}^{K-1}\mathbb{E}\left[\|\mathfrak{g}(x_k, y_k)\|^2\right] \leq \mathcal{O}\left(\frac{N}{K} + \sqrt{\frac{N}{K}}\right)$, which leads to

$$\frac{1}{K}\sum_{k=0}^{K-1}\mathbb{E}\left[\|\mathfrak{g}(x_k, y_k)\|\right] \leq \mathcal{O}\left(\sqrt{\frac{N}{K} + \sqrt{\frac{N}{K}}}\right) = \mathcal{O}\left(\left(\frac{N}{K}\right)^{\frac{1}{4}}\right).$$

In the second case, we have

$$\frac{1}{16\alpha}\|x_{k+1} - x_k\|^2 + \frac{1}{16\beta}\|y_k - y_+(z_k)\|^2 + \frac{p}{16\gamma}\|z_{k+1} - z_k\|^2$$

$$\geq 24p\gamma\|x^*(z_k) - x(y_+(z_k), z_k)\|^2.$$

Then, according to (21), we have

$$\mathbb{E}\left[\phi_k - \phi_{k+1}\right]$$

$$\geq \mathbb{E}\left[\frac{1}{16\alpha}\|x_{k+1} - x_k\|^2 + \frac{1}{16\beta}\|y_k - y_+(z_k)\|^2 + \frac{p\gamma}{16}\|x_{k+1} - z_k\|^2\right]$$

$$- \frac{L^3 r_k^2 \alpha^2 d_x}{N} - 12\beta^2 L^2 r_k^2 \sigma_2^2 p\gamma d_x - \frac{\beta L^2 r_k^2 d_x}{8} - \frac{L^2 r_k^2 d_x}{8N}$$

$$\geq t_2 \mathbb{E}\left[\|\mathfrak{g}(x_k, y_k)\|^2\right] - \frac{5t_2 L^2 r_k^2 d_x}{4} - \frac{L^3 r_k^2 \alpha^2 d_x}{N} - 12\beta^2 L^2 r_k^2 \sigma_2^2 d_x - \frac{\beta L^2 r_k^2 d_x}{8} - \frac{L^2 r_k^2 d_x}{8N},$$

where $t_2 = \min\left\{\frac{1}{16\alpha\left(\frac{3N}{\alpha^2} + 8L^2\sigma_3^2 + 6p^2\right)}, \frac{\beta}{32}, \frac{\gamma}{96p}\right\}$. Consequently, we can obtain

$$\mathbb{E}\left[\|\mathfrak{g}(x_k, y_k)\|^2\right] \leq \frac{\mathbb{E}\left[\phi_k - \phi_{k+1}\right]}{t_2} + \frac{5L^2 r_k^2 d_x}{4} + \frac{L^3 r_k^2 \alpha^2 d_x}{Nt_2}$$

$$+ \frac{12\beta^2 L^2 r_k^2 \sigma_2^2 d_x}{t_2} + \frac{\beta L^2 r_k^2 d_x}{8t_2} + \frac{L^2 r_k^2 d_x}{8Nt_2}.$$

Taking the summation of the above inequality from $k = 0$ to $K - 1$ leads to

$$\frac{1}{K}\sum_{k=0}^{K-1} \mathbb{E}\left[\|\mathfrak{g}(x_k, y_k)\|^2\right]$$

$$\leq \frac{\phi_0 - f}{Kt_2} + \frac{5L^2 d_x \sum_{k=0}^{K-1} r_k^2}{4K} + \frac{L^3 \alpha^2 d_x \sum_{k=0}^{K-1} r_k^2}{Nt_2 K}$$

$$+ \frac{12\beta^2 L^2 \sigma_2^2 p\gamma d_x \sum_{k=0}^{K-1} r_k^2}{t_2 K} + \frac{\beta L^2 d_x \sum_{k=0}^{K-1} r_k^2}{8t_2 K} + \frac{L^2 d_x \sum_{k=0}^{K-1} r_k^2}{8Nt_2 K}.$$

Substituting $\gamma = \frac{1}{\sqrt{KN}}$ and $\sum_{k=0}^{K-1} r_k^2 \leq \frac{1}{b}$ into the above inequality, we can get

$$\frac{1}{K}\sum_{k=0}^{K-1} \mathbb{E}\left[\|\mathfrak{g}(x_k, y_k)\|^2\right] \leq \mathcal{O}\left(\sqrt{\frac{N}{K}}\right).$$

Finally, we have

$$\min_{k=0,\cdots,K-1} \mathbb{E}\left[\|\mathfrak{g}(x_k, y_k)\|\right] \leq \frac{1}{K}\sum_{k=0}^{K-1} \mathbb{E}\left[\|\mathfrak{g}(x_k, y_k)\|\right] \leq \mathcal{O}\left(\left(\frac{N}{K}\right)^{\frac{1}{4}}\right),$$

which is the desired result and finishes the proof.

## D   PROOF OF LEMMA 5.1

We provide the definition of a critical KKT point below.

**Definition D.1.** We say a point $x^*$ is a critical KKT point of problem (1) if $c_j(x^*) \leq 0, \forall j \in \mathcal{J}$ and there exists $y^* \in \mathcal{Y}$ satisfying $y^* \geq 0$ and

$$y^*(j)c_j(x^*) = 0, \forall j \in \mathcal{J}, \tag{30}$$

$$\nabla_x h(x^*) + (y^*)^T \nabla_x c(x^*) = 0. \tag{31}$$

We restate Lemma 5.1 as follows.

**Lemma D.1.** *Suppose that $(x, y) \in \mathbb{R}^{d_x} \times \mathcal{Y}$ is a stationery point of $f(x, y)$ satisfying $\|\mathfrak{g}(x, y)\| = 0$ and $y < \overline{y}$. Then, $x$ is a critical KKT point of problem (1).*

*Proof of Lemma 5.1.* Recall that $\|\mathfrak{g}(x, y)\| = 0$ for some $(x, y) \in \mathbb{R}^{d_x} \times \mathcal{Y}$ implies that

$$
\begin{pmatrix} \mathfrak{g}_x(x, y) \\ \mathfrak{g}_y(x, y) \end{pmatrix} = \begin{pmatrix} \nabla_x f(x, y) \\ (1/\beta)\left(y - \mathcal{P}_\mathcal{Y}\left[y + \beta \nabla_y f(x, y)\right]\right) \end{pmatrix}
$$

$$
= \begin{pmatrix} \nabla_x h(x) + y^T \nabla_x c(x) \\ (1/\beta)\left(y - \mathcal{P}_\mathcal{Y}\left[y + \beta c(x)\right]\right) \end{pmatrix}
$$

$$
= 0.
$$

The first condition $\nabla_x h(x) + y^T \nabla_x c(x) = 0$ implies (31) directly. The second condition implies that $y - \mathcal{P}_\mathcal{Y}\left[y + \beta c(x)\right] = 0$. Then, due to that $y < \overline{y}$, one of the following two cases holds: (1) $c(x) = 0$; (2) $y = 0$ and $c(x) \leq 0$. Both cases can lead to $c(x) \leq 0$ and (30).

$\square$

# E   DETAILED EXPERIMENTAL SETTINGS

## E.1   PROBLEM FORMULATION

We consider a classic energy management problem in power systems called load curtailment. In this problem, a load aggregator tries to coordinate the loads of multiple users within a distribution network to meet the load requirements imposed by the higher-level grid operator. On the one hand, the aggregator needs to ensure that the total power injection into the network satisfies a constraint tied to a reference load. On the other hand, the operational costs associated with users' load adjustments should be minimized to maintain a satisfactory consumer experience.

Mathematically, denote $x \in \mathbb{R}^{d_x}$ as the power load of multiple users. Let $D$ denote the reference load received from the grid operator, which imposes a constraint on the distribution network's net power exchange with the main grid. The power injection of the distribution network is not simply the sum of users' loads but is determined by nonlinear power flow dynamics. Denote the dynamics as a function of load levels of multiple users $p_c(x) : \mathbb{R}^{d_x} \to \mathbb{R}$. In our formulation, $p_c(x)$ is viewed as a black box, provided that the topology and parameters of the distribution network are unattainable. That means we can only observe the total power consumption $p_c(x)$ of a distribution network given the power load $x$ of users.

We apply the 141-bus distribution network model as the nonconvex black box (shown in Figure 3) (Khodr et al., 2008). We consider the following problem with $d_x = 168$:

$$
\min_{x \in \mathbb{R}^{168}} h(x) = \sum_{i \in [168]} c_i(x(i)) + \rho(x),
$$

$$
s.t.\ p_c(x) \leq D,\ x \in \mathcal{X} = [\underline{x}, \overline{x}],
$$

(32)

where $c_i : \mathbb{R} \to \mathbb{R}$ is the cost function of user $i$ and defined as $c_i(x(i)) = a_i x^2(i) + b_i x(i)$. $\rho(x) : \mathcal{X} \to \mathbb{R}$ is the penalty term when the voltage is out of the standard region and formulated as

$$
\rho(x) = \sum_{j \in [141]} \left(\max(v_j(x) - \overline{v}, 0)^2 + \max(\underline{v} - v_j(x), 0)^2\right).
$$

Here $v_j(x)$ denotes the voltage of the $j$th node when the power load of the distribution network is $x$, and $\underline{v}, \overline{v}$ represent the lower and upper bounds of the voltage. $v_j : \mathbb{R}^{168} \to \mathbb{R}, \forall j$ is also a black-box mapping with only the function value observable. Therefore, the objective and constraint functions in problem (32) are both non-analytical.

## E.2   PARAMETERS SETTING

In our numerical simulation, $\underline{x}$ is also set to 0, and $\overline{x}$ is the nominated load level from the original 141-bus system. The coefficients of cost functions, $a_i$ and $b_i$, are randomly sampled from the

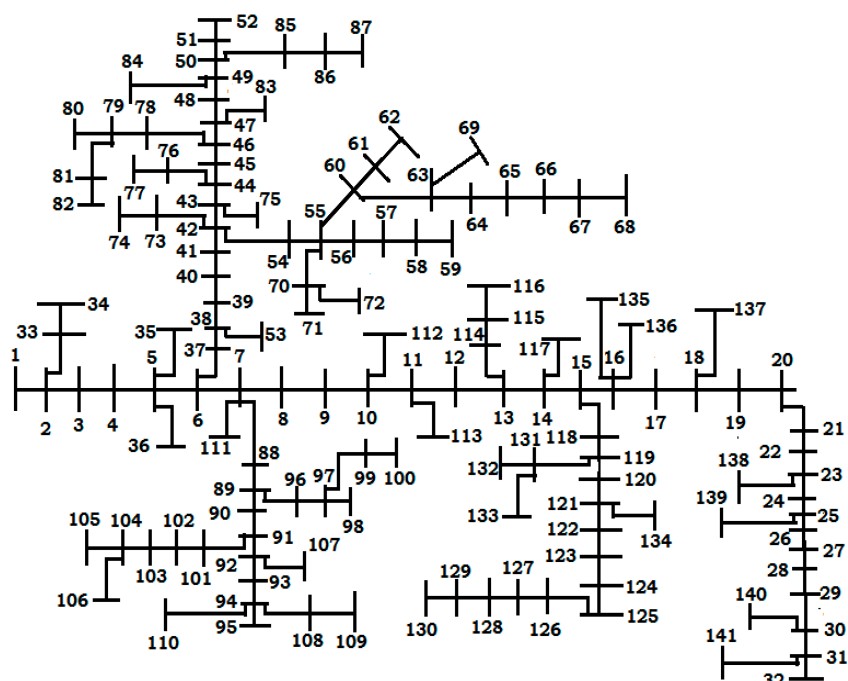

Figure 3: 141-bus distribution network.

intervals $(0.5, 1.5)$ and $(0, 5)$, respectively. For the voltage penalty, we set $\underline{v} = 0.96$ $p.u.$ and $\overline{v} = 1.04$ $p.u..$ The total load to be curtailed is set to $0.15$ $p.u. = 1500$ $kW$. In addition to testing our proposed algorithms, we also compare them with three other algorithms, ZO-MinMax (Liu et al., 2020b), ZO-AGP (Xu et al., 2024), and stochastic zeroth-order constraint extrapolation (SZO-ConEX) (Nguyen & Balasubramanian, 2023). For all tests, we compute $\|\mathfrak{g}(x_k, y_k)\|$ as the stationarity measures. The constraint violation is measured by $p_c(x) - D$. In Table 2, the relative error is computed by $(h(x_k) - h^*)/h^*$ based on the optimal objective function value $h^* = 0.3841$.

For ZOB-GDA, we consider four scenarios with batch sizes $b = 1, 10, 50, 168$, where the step sizes are set as $\alpha = 0.03, 0.025, 0.005, 0.00035$ and $\beta = 0.01\alpha$, respectively. For ZOB-SGDA, we set $p = 10$ and $\gamma = 0.3$, and adopt the same batch-size scenarios and corresponding step sizes as ZOB-GDA. For the benchmark algorithms, ZO-MinMax is implemented with $\alpha = \beta = 5 \times 10^{-6}$; since it involves constraint-handling techniques, we adopt a decaying penalty parameter $\delta_k = \min(50/k, 0.1)$. For SZO-ConEx, we set $\alpha = \beta = 5 \times 10^{-6}$. For ZOAGP, we choose $\alpha = \beta = 0.00035$. We set the maximum iteration steps as $K = 20000$. For the smoothing radius, ZOB-GDA, ZOB-SGDA, and ZOAGP use $r_k = \min(10^{-1}/(k^{1.2}), 2 \times 10^{-4})$, while SZO-ConEx and ZO-MinMax use $r_k = \min(10^{-2}/(k + 4000)^{1.1}, 1 \times 10^{-5})$ All the parameters are selected with the best performance among multiple settings. All experiments are conducted on a MacBook Pro laptop equipped with an Apple M1 Pro SoC (10-core CPU: 8 performance cores and 2 efficiency cores) and 16 GB of unified memory.

### E.3 NUMERICAL RESULTS FOR NOISY CASES

In this subsection, we further test our proposed algorithms, ZOB-GDA and ZOB-SGDA, under noisy observations to validate their robustness. The observed values of $p_c(x)$ are perturbed by additive Gaussian noise with zero mean and a standard deviation of 5 kW. In noisy cases, the smoothing radius is set as $r_k = \min(400/(k^{1.2}), 4 \times 10^{-3})$, while all the other parameter settings remain unchanged as in the noise-free cases. Each algorithm is tested with 20 runs, and the average performance is presented in Figure 4. The average number of queries required to generate solutions with different qualities is summarized in Table 3. The results show that our algorithms also exhibit satisfactory performance in noisy cases without degrading significantly, which demonstrates the robustness of our methods.

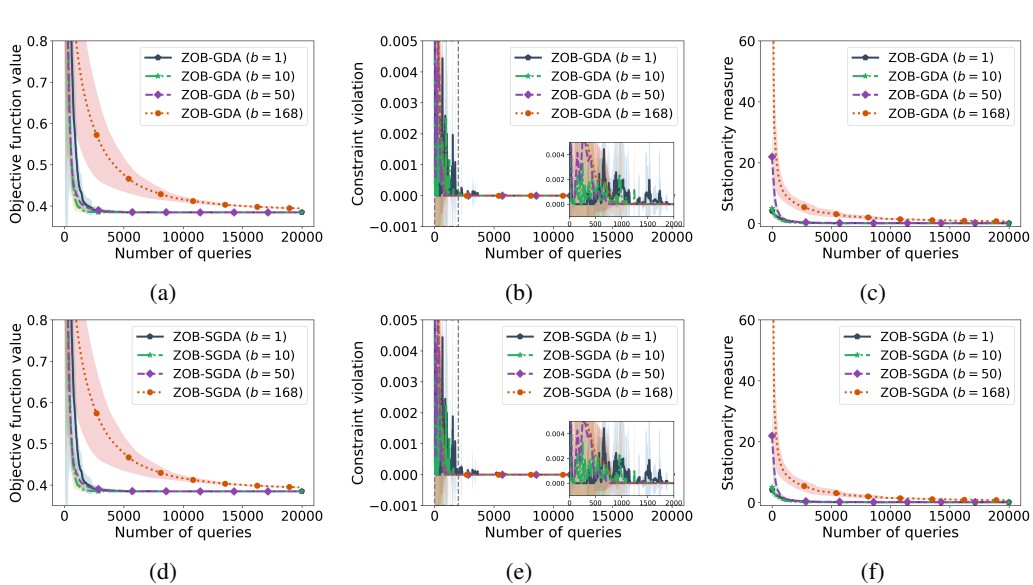

Figure 4: Performance of ZOB-GDA and ZOB-SGDA under noisy observations. (a), (b), and (c) present the objective function value, constraint violation, and stationarity measure of ZOB-GDA. (d), (e), and (f) present the corresponding results for ZOB-SGDA.

Table 3: Average numbers of iterations and queries required to generate solutions with certain levels of relative errors and zero constraint violation under noisy observation.

| Relative error | | 10% | | 1% | | 0.1% | |
|---|---|---|---|---|---|---|---|
| | | Iteration | Complexity | Iteration | Complexity | Iteration | Complexity |
| ZOBGDA | b=1 | 680.8 | 1361.6 | 1263.3 | 2526.6 | 1816.75 | 3633.5 |
| | b=10 | 75.05 | 825.55 | 139.25 | 1531.75 | 194.55 | 2140.05 |
| | b=50 | 21.65 | 1104.15 | 60.65 | 3093.15 | 97.05 | 4949.55 |
| | b=168 | 51.9 | 8771.1 | 194.8 | 32921.2 | 255.25 | 43137.25 |
| ZOB-SGDA | b=1 | 687.4 | 1374.8 | 1282.15 | 2564.3 | 1937.35 | 3874.7 |
| | b=10 | 76.95 | 846.45 | 141 | 1551 | 191.05 | 2101.55 |
| | b=50 | 21.8 | 1111.8 | 61.6 | 3141.6 | 101.2 | 5161.2 |
| | b=168 | 52.25 | 8830.25 | 196.65 | 33233.85 | 257.25 | 43475.25 |

