# OpenReview forum: "Query-Efficient Zeroth-Order Algorithms for Nonconvex Optimization"
_ICLR.cc/2026/Conference — ICLR 2026 Conference Withdrawn Submission_

### Official Review · Reviewer_mdgC · 2025-10-28

**Soundness:** 2
**Presentation:** 3
**Contribution:** 2
**Rating:** 4
**Confidence:** 3

**Summary:**

The paper tackles the problem of reducing per-step query complexity while maintaining overall query efficiency in zeroth-order (gradient-free) optimization for nonconvex, constrained black-box problems. The authors introduce two algorithms—ZOB-GDA and ZOB-SGDA—which embed block-coordinate gradient estimators within smoothed gradient descent–ascent (GDA) frameworks. By updating only a randomly selected block of variables at each iteration, the methods achieve flexible per-step query costs O(b) for block size b and demonstrate state-of-the-art overall query complexity for finding stationary points. Theoretical convergence guarantees are provided, and experiments on energy management tasks validate the query efficiency against recent baselines.

**Strengths:**

1. The work tackles a clear limitation in zeroth-order optimization—balancing per-step efficiency and overall convergence, particularly for constrained, nonconvex problems where standard RGE or CGE methods are either too slow per iteration or have high total query cost (Sections 1, 2.2).

2. The paper rigorously proves finite-sample convergence for both ZOB-GDA and ZOB-SGDA, including explicit complexity bounds. In particular, Theorem 4.1 and Corollary 4.2 for ZOB-SGDA yield a query complexity matching the best-known results, and Section 3.2 provides a detailed derivation for ZOB-GDA.

**Weaknesses:**

1. Experimental Validation. The numerical experiments (Section 6, Figures 1 and 2) are limited to a single energy management application where $d_x$ is only 168. Broader empirical evaluation—spanning more domains and higher-dimensional and/or more complex constraint sets—would underscore generalizability. The per-iteration complexity is a significant challenging only/especially when $d_x$ is large enough.

2. Claims such as “over 10 times fewer queries” (Abstract, Section 6, Table 2) merit more nuanced discussion—performance benefits are shown on a single problem and are dependent on hyperparameters (e.g., choice of block size $b$), so the result may not transfer universally.

3. Some theoretical choice and analysis needs some more intuition to understand better. For example, in Section 3.2,  what are the intuitions that bias can be bounded and how is that related to variance?

4. (Minor) Notation $K$ has been abused such as in Algorithm 2 to denote both number of iterations and smoothed function.

**Questions:**

See above.

---

### Official Review · Reviewer_X7tH · 2025-10-31

**Soundness:** 3
**Presentation:** 3
**Contribution:** 3
**Rating:** 4
**Confidence:** 3

**Summary:**

This paper addresses zeroth-order (ZO) optimization for constrained black-box problems where both objective and constraint functions lack analytical expressions. The authors propose two novel algorithms: ZOB-GDA (Zeroth-Order Block Gradient Descent Ascent) and ZOB-SGDA (Zeroth-Order Block Smoothed Gradient Descent Ascent). The key innovation is integrating block coordinate updates with gradient descent-ascent methods, which allows for adjustable single-step query efficiency while maintaining optimal overall query complexity. Specifically, ZOB-SGDA achieves optimal $O(d/\epsilon^4)$ overall query complexity while requiring only $O(b)$ queries per step, compared to $d$ for traditional coordinate-wise gradient estimation methods. The authors provide finite-sample convergence guarantees for nonconvex-concave minimax problems and demonstrate superior empirical performance on a power system load curtailment problem.

**Strengths:**

1. The paper makes an important observation about the trade-off between single-step efficiency and overall query complexity in zeroth-order optimization.

2. The convergence analysis is rigorous and comprehensive, establishing finite-sample guarantees for nonconvex-concave problems. ZOB-SGDA achieves the best-known query complexity $O(d/\epsilon^4)$, matching optimal results while offering flexiable single-step efficiency.

3.  The paper is generally well-written with clear motivation. The comparison table (Table 1) effectively highlights the contributions relative to existing work.

**Weaknesses:**

1. Experiments focus on a single power-systems task at $d=168$. Given the paper’s relevance to ML-style black-box tasks, additional benchmarks (e.g., adversarial attacks, policy optimization, or LLM-related ZO tasks) would strengthen external validity and showcase scalability to high dimensional problems.

2. No comparison of wall-clock time, which is important for assessing practical efficiency.

3. The block size b is a key feature, allowing for single-step query costs from 1 to d. The empirical results in Table 2  are very interesting, as they show that $b=10$ is the most query-efficient, outperforming both $b=1$ and the full-batch $b=168$. This suggests a non-trivial trade-off. However, the paper provides no discussion or heuristics on how to set this crucial hyperparameter in practice.

4. The current title, “Query-Efficient Zeroth-Order Algorithms for Nonconvex Optimization,” is broad and doesn’t signal that the paper tackles constrained black-box problems.

**Questions:**

1. Can you provide principled guidelines or an adaptive strategy for choosing block size b? How does the optimal b depend on problem dimension or computational budget? If it possible to develop adaptive block size b across iterations?

2. The paper emphasizes query complexity; could you also report wall-clock time with/without parallelizing coordinate/block queries, to show real-time benefits in practical deployments?

---

### Official Review · Reviewer_hA6b · 2025-11-01

**Soundness:** 3
**Presentation:** 3
**Contribution:** 3
**Rating:** 4
**Confidence:** 3

**Summary:**

In this paper, the authors studied query-efficient zeroth-order optimization for constrained nonconvex problems. They first transformed the original formulation into a minimax problem, and proposed two algorithms, ZOB-GDA and ZOB-SGDA, which combine random block updates with gradient descent–ascent (GDA) to estimate gradients using only a subset of coordinates at each iteration instead of all dimensions. By integrating this block-based estimator into primal–dual updates, the methods achieve overall query complexity of $\mathcal{O}(d\epsilon^{-4})$ for finding an $\epsilon$-stationary point, which matches the best-known overall query complexity. The authors provide numerical evidence on a practical problem where the proposed methods require less function queries than prior ZO algorithms when using a block size between 1 and model dimension.

**Strengths:**

- The idea of leveraging block coordinate updates in nonconvex-concave minimax optimization is novel.
- The empirical results are promising when varying the block size.

**Weaknesses:**

- The setting in this paper is limited. They consider the minimax problem (2) that is linear in $y$. Also, the methods only consider the deterministic case.
- The theoretical analysis, though technically sound, is incremental. The overall query complexity matches the best-known result, but does not improve upon it.
- Some experiment results do not align well with theory (see questions below).

**Questions:**

- Questions
  - In Definition 2.1, why is the stationarity measure defined based on the two metrics, instead of one? It seems the convergence is only established based on individual metrics.
  - In Table 2, the result of SZO-ConEx and that of ZOB-GDA (b = 1) have a large gap. What is the explanation for this? In theory, both methods have the same complexity in terms of queries per step and overall queries.
  - In Table 2, why is ZOB-SGDA worse than ZOB-GDA? In theory, ZOB-SGDA should have much better overall query complexity.
- Typos
  - Line 217: k-th iterate -> k-th iteration

---

### Official Review · Reviewer_3ivC · 2025-11-03

**Soundness:** 3
**Presentation:** 3
**Contribution:** 2
**Rating:** 4
**Confidence:** 3

**Summary:**

This paper studies the problem of zeroth-order constrained optimization (1) in the nonconvex–concave setting (2). The authors propose two new algorithms, ZOB-GDA and ZOB-SGDA, by integrating block coordinate updates into the zeroth-order gradient descent-ascent framework. The proposed methods aim to improve single-step query complexity without worsening the overall query complexity. Theoretical convergence guarantees and numerical experiments are provided.

**Strengths:**

1. The paper is overall well-organized and clearly written. Assumptions are explicitly stated, theorems are formally presented.

2. Theoretical results are technically sound. Although I didn't check the proof carefully, the theorems seem to be right.

**Weaknesses:**

1. My main concern is the motivation for introducing a block-coordinate algorithm in this problem and query setting. The paper does not clearly explain why block-coordinate updates are necessary or beneficial here. Theoretical results show that the overall query complexity of the proposed methods matches, but does not improve upon, existing results. Moreover, the proposed algorithms have higher per-iteration computational costs than randomized gradient estimation (RGE) methods and only beat coordinate-wise gradient estimation (CGE) in this aspect.
On the other hand, the benefits of lower per-iteration query complexity are unclear. With sufficient computational resources, queries within each iteration can be parallelized, in which case methods such as ZOAGP could achieve lower wall-clock runtime. In fact, higher per-iteration query complexity is not necessarily a drawback in practice.
While I acknowledge that ZOB-GDA generalizes ZOAGP as a special case when $b=1$ and provide a compromised option for larger $b$, this extension alone does not seem substantial enough to publish at ICLR without a stronger theoretical or practical justification.

2. The empirical results show some performance gains for the proposed algorithms on specific examples, though these advantages are not reflected in the theoretical analysis. I find the experiments unconvincing for two reasons:
First, the 141-bus distribution network example appears to be a hand-picked application that is unlikely to arise naturally in a zeroth-order optimization context. It gives the impression of being chosen to favor the proposed method.
Second, the experimental evaluation is too narrow. More standard constrained optimization benchmarks should be included to draw reliable empirical conclusions.

3. The discussion in Section 5, at least the first half, reiterates well-known results from basic optimization theory. These points are standard material in even undergraduate optimization courses and do not add meaningful insights to the paper. I recommend substantially condensing or removing this part.

**Questions:**

NA

---

### Official Review · Reviewer_3GP5 · 2025-11-04

**Soundness:** 2
**Presentation:** 2
**Contribution:** 2
**Rating:** 4
**Confidence:** 3

**Summary:**

This paper proposes two zeroth-order optimization algorithms (i.e., ZOB-GDA and ZOB-SGDA) by integrating block updates with gradient descent ascent (GDA) and smoothed GDA (SGDA). The proposed algorithms can simultaneously achieve efficiency in single-step gradient estimation and overall query complexity.

**Strengths:**

Strength:

1. The paper provides comprehensive finite-sample convergence analyses for both algorithms.

2. For ZOB-SGDA, the authors establish an overall query complexity bound, which matches the state-of-the-art result for nonconvex-concave minimax problems, while ZOB-GDA achieves a reasonable trade-off for its simplicity.

**Weaknesses:**

Weakness:

1. The paper notes in Section 5 that equality constraints can be converted to two inequalities but provides no theoretical or empirical validation of this approach.

2.The block size is shown to be a practical tuning parameter. How should the block size be chosen for a given problem? For example, is there a heuristic to balance single-step queries and the number of blocks, which influences convergence rate via Theorem 3.1?

3. This paper provides no details on the sampling distribution of the blocks.

4. For non-uniform sampling (e.g., weighting blocks by gradient magnitude), could the algorithm achieve faster convergence?

5. How sensitive are the algorithms to the smoothing radius and step sizes?

6. The experimental results are not convincing. For instance, the authors should compare the performance of the two proposed algorithms, ZOB-GDA and ZOB-SGDA.

**Questions:**

1. The paper notes in Section 5 that equality constraints can be converted to two inequalities but provides no theoretical or empirical validation of this approach.

2.The block size is shown to be a practical tuning parameter. How should the block size be chosen for a given problem? For example, is there a heuristic to balance single-step queries and the number of blocks, which influences convergence rate via Theorem 3.1?

3. This paper provides no details on the sampling distribution of the blocks.

4. For non-uniform sampling (e.g., weighting blocks by gradient magnitude), could the algorithm achieve faster convergence?

5. How sensitive are the algorithms to the smoothing radius and step sizes?

6. The experimental results are not convincing. For instance, the authors should compare the performance of the two proposed algorithms, ZOB-GDA and ZOB-SGDA.

---

### Note · Authors · 2025-12-03

I have read and agree with the venue's withdrawal policy on behalf of myself and my co-authors.